# The end-joining factor Ku acts in the end-resection of double strand break-free arrested replication forks

Ana Teixeira-Silva [1,2], Anissia Ait Saada[1,2], Julien Hardy[1,2], Ismail Iraqui[1,2], Marina Charlotte Nocente[1,2], Karine Fréon[1,2] & Sarah A.E. Lambert[1,2]

Replication requires homologous recombination (HR) to stabilize and restart terminally arrested forks. HR-mediated fork processing requires single stranded DNA (ssDNA) gaps and not necessarily double strand breaks. We used genetic and molecular assays to investigate fork-resection and restart at dysfunctional, unbroken forks in *Schizosaccharomyces pombe*. Here, we report that fork-resection is a two-step process regulated by the non-homologous end joining factor Ku. An initial resection mediated by MRN-Ctp1 removes Ku from terminally arrested forks, generating ~110 bp sized gaps obligatory for subsequent Exo1-mediated long-range resection and replication restart. The mere lack of Ku impacts the processing of arrested forks, leading to an extensive resection, a reduced recruitment of RPA and Rad51 and a slower fork-restart process. We propose that terminally arrested forks undergo fork reversal, providing a single DNA end for Ku binding. We uncover a role for Ku in regulating end-resection of unbroken forks and in fine-tuning HR-mediated replication restart.

---

[1] Institut Curie, PSL Research University, CNRS, UMR3348, Orsay F-91405, France. [2] University Paris Sud, Paris-Saclay University, CNRS, UMR3348, Orsay F-91405, France. Anissia Ait Saada and Julien Hardy contributed equally to this work. Correspondence and requests for materials should be addressed to S.A.E.L. (email: sarah.lambert@curie.fr)

At each cell division, ensuring the correct duplication and segregation of the genetic material is crucial for maintaining genome integrity. Although mutational events contribute to genome evolution, DNA lesions trigger genome instability often associated with human diseases such as cancer, genomic disorders, aging and neurological dysfunctions[1].

DNA replication stress constitutes a major peril for genome stability. This is particularly evident in the case of oncogene-induced proliferation, which results in replication stress and faulty genome duplication, contributing to acquire genetic instability in neoplasic lesions[2,3]. A key feature of replication stress is the alteration of replication fork progression by numerous replication fork barriers (RFBs), DNA damage, clashes between transcription and replication machineries, DNA secondary structures and unbalanced dNTP pools. Replication stress threatens faithful DNA duplication[4]. Fork obstacles impact replisomes' functionality, and may result in replication forks stalling, requiring stabilization by the S-phase checkpoint to ensure DNA synthesis resumption[5]. Fork obstacles may also result in dysfunctional and terminally arrested forks, which lack their replication-competent state, and necessitate additional mechanisms to resume DNA synthesis. Through nucleolytic cleavage, terminally arrested forks are converted into broken forks, exhibiting one ended DSB[6]. A DNA nick directly converts an active fork into a broken fork, accompanied with a loss of some replisome components[7]. Forks lacking their replication competence, and thus terminally arrested, are often referred to as collapsed forks, whether broken or not.

Homologous recombination (HR) is one of the pathways involved in counteracting the deleterious outcomes of replication stress by ensuring the repair of DSBs and securing DNA replication[8]. HR is initiated by the loading of the recombinase Rad51 onto ssDNA, with the assistance of mediators such as yeast Rad52 and mammalian BRCA2. The Rad51 filament then promotes homology search and strand invasion with an intact homologous DNA template. HR allows the repair of forks exhibiting one ended DSB likely through break induced replication[9–12]. However, growing evidences point out that DSB, are not a pre-requirement neither for the recruitment of HR factors at dysfunctional forks nor for their restart[13–17].

DSBs are also repaired by the non-homologous end joining (NHEJ) pathway which promotes the direct ligation of DNA ends with limited or no end-resection[18]. While HR is active in S and G2-phase, NHEJ is active throughout the cell cycle. A key component of NHEJ is the heterodimer Ku composed of two subunits, Ku70 and Ku80, both required for the stability of the complex. Yeast Ku binds dsDNA ends, inhibits end-resection, and allows the ligation of DSBs trough the recruitment of Ligase 4[19–21]. Interestingly, Ku is also involved in the repair of replication-born DSBs, where it limits end-resection[21–24] and yeast Ku acts as a backup to promote cell survival upon replication stress[25,26].

In most eukaryotes, DSB resection is a two-step process[27,28]. An initial 5′ to 3′ nucleolytic processing, limited to the vicinity of the DNA end, is mediated by the MRN (Mre11-Rad50-Nbs1) complex which binds DSBs as an early sensor[29]. MRN then recruits Ctp1, a protein reported to share a conserved C-terminal domain with the mammalian nuclease CtIP and with *Saccharomyces cerevisiae* Sae2[30,31]. The endo- and exo-nuclease activities of Mre11, stimulated by Sae2, are not strictly required to DNA end resection at "clean" DSBs, but are critical to process "dirty" ends (generated by gamma-irradiation) or blocked ends such as meiotic DSBs at which Spo11 remains covalently attached to the DNA[19,32–35]. In yeasts, the nuclease activity of Mre11 and the Ctp1-dependent clipping function are required for Ku removal from DSBs[19,36–38]. The second DNA end-resection step consists of a 5′ to 3′ long-range resection mediated by two conserved[33–35],

but separate pathways dependent on either the exonuclease Exo1 or the helicase-nuclease Sgs1 (the *S. pombe* Rqh1 and the mammalian BLM orthologue)-Dna2[27,28]. The long-range resection creates a longer 3′ tailed ssDNA coated by RPA, up to 2–4 kb[29].

The nucleolytic processing of nascent strands at stalled replication forks is central to resume DNA synthesis but uncontrolled resection is detrimental to genome stability[39]. There is a growing interest in understanding how ssDNA forms at forks since it's constitutes a key activator of the DNA replication checkpoint, an anti-cancer therapeutic target[40]. Also, fork-degradation prevented by BRCA2 plays a pivotal role in the chemo-resistance of breast cancer cell lines[41]. Many of the factors required for DSBs resection are also involved in the end-resection of replication forks, such as MRN, DNA2 and CtIP[13,15,42–44]. However, contrary to DSBs, the orchestration of fork-resection is poorly understood.

Here, we investigated the formation of ssDNA gaps using a model of terminally and DSB-free arrested forks. Resection of newly replicated strands occurs in two steps. An initial resection mediated by MRN-Ctp1 generates short ssDNA gaps of ~110 bp in size which are obligatory to promote Rad51-mediated fork-restart. A long-range resection mediated by Exo1, but not Rqh1, generates larger ssDNA gaps which are dispensable to replication restart. Unexpectedly, despite the absence of DSB, we found that the MRN-Ctp1 pathway removes Ku from terminally arrested forks, allowing long-range resection to occur. The mere lack of Ku results in an extensive fork-resection, a reduced amount of RPA and Rad51 recruited to arrested forks and in slower replication fork-restart. We propose that Ku regulates the resection of DSB-free arrested forks by ensuring that fork-resection occurs as a two-step process. Our data are consistent with dysfunctional forks undergoing fork reversal, which provides a single DNA end for Ku binding. We uncover a role for Ku in regulating the resection of terminally arrested forks, in the absence of DSB, and in fine-tuning replication restart.

## Results

**Conditional RFBs to monitor fork-resection and restart**. We exploited a previously described conditional replication fork barrier (RFB), named *RTS1*, to block a replisome in a polar manner at a specific locus[45] (Fig. 1a). The blockage of the replication fork is mediated by the *RTS1*-bound protein Rtf1 whose expression is controlled through the *nmt41* promoter repressed in the presence of thiamine. Upon Rtf1 expression, > 90% of forks travelling in the main replication direction are blocked at the *RTS1*-RFB resulting in dysfunctional forks. These terminally arrested forks are either restarted by HR or rescued by the progression of opposite forks[17]. HR-mediated replication restart occurs in 20 min and is initiated by an ssDNA gap, not a DSB, onto which RPA, Rad52 and Rad51 are loaded[46–48].

Restarted forks are associated with a non-processive DNA synthesis, liable to replication slippage (RS)[46,49]. Both strands of restarted forks are synthetized by Polymerase delta, reflecting a non-canonical replisome likely insensitive to the RFB[50]. We have previously developed a reporter assay consisting of an inactivated allele of *ura4*, *ura4-sd20*, which allows us to infer the degree of RS caused by the restarted fork by monitoring the frequency of Ura+ reversion upon expression of Rtf1 (Supplementary Fig. 1). The frequency of Ura+ reversion is used as readout of the frequency at which the *ura4-sd20* allele is replicated by a restarted fork in the cell population[49]. The *ura4-sd20* allele was inserted either downstream or upstream from the *RTS1*-RFB to generate the construct *t-ura4sd20 < ori* or *t < ura4sd20-ori* (*t* for telomere, < for the *RTS1*-RFB and its polarity)*, respectively (Fig. 1b). To monitor the basal level of RS in different genetic backgrounds,

we also generated a *t-ura4sd20-ori* construct, devoid of the *RTS1*-RFB. To obtain the true occurrence of RS by the *RTS1*-RFB, independently of the genetic background, we subtracted the RS frequency of the strain devoid of RFB from the frequency of the strain containing the *t-ura4sd20 < ori* construct, upon expression of Rtf1 (Supplementary Fig. 1). We showed that RS occurring upstream and downstream from the *RTS1*-RFB are both consequences of restarted forks and dependent on Rad51 and Rad52 (Supplementary Fig. 1)[49]. Indeed, DNA synthesis associated with Rad51-mediated fork-restart occurs occasionally upstream from the initial site of fork arrest, as a consequence of ssDNA gap formation by Exo1[47]. Thus, the RFB-induced Ura⁺

reversion assay allows the quantification of replication restart efficiency and an indirect monitoring of fork-resection.

**A two-step resection occurs at terminally arrested forks.** HR-mediated fork-restart at the *RTS1*-RFB is initiated by ssDNA gaps of ~ 1 kb in size formed upstream from terminally arrested forks[47]. Indeed, no site-specific DSBs were detectable at the active *RTS1*-RFB by pulsed field gel electrophoresis and Southern-blot, even in the absence of HR[16,17]. Also, the analysis of recombination intermediates at the *RTS1*-RFB was consistent with fork-restart occurring through a template switch event initiated by a

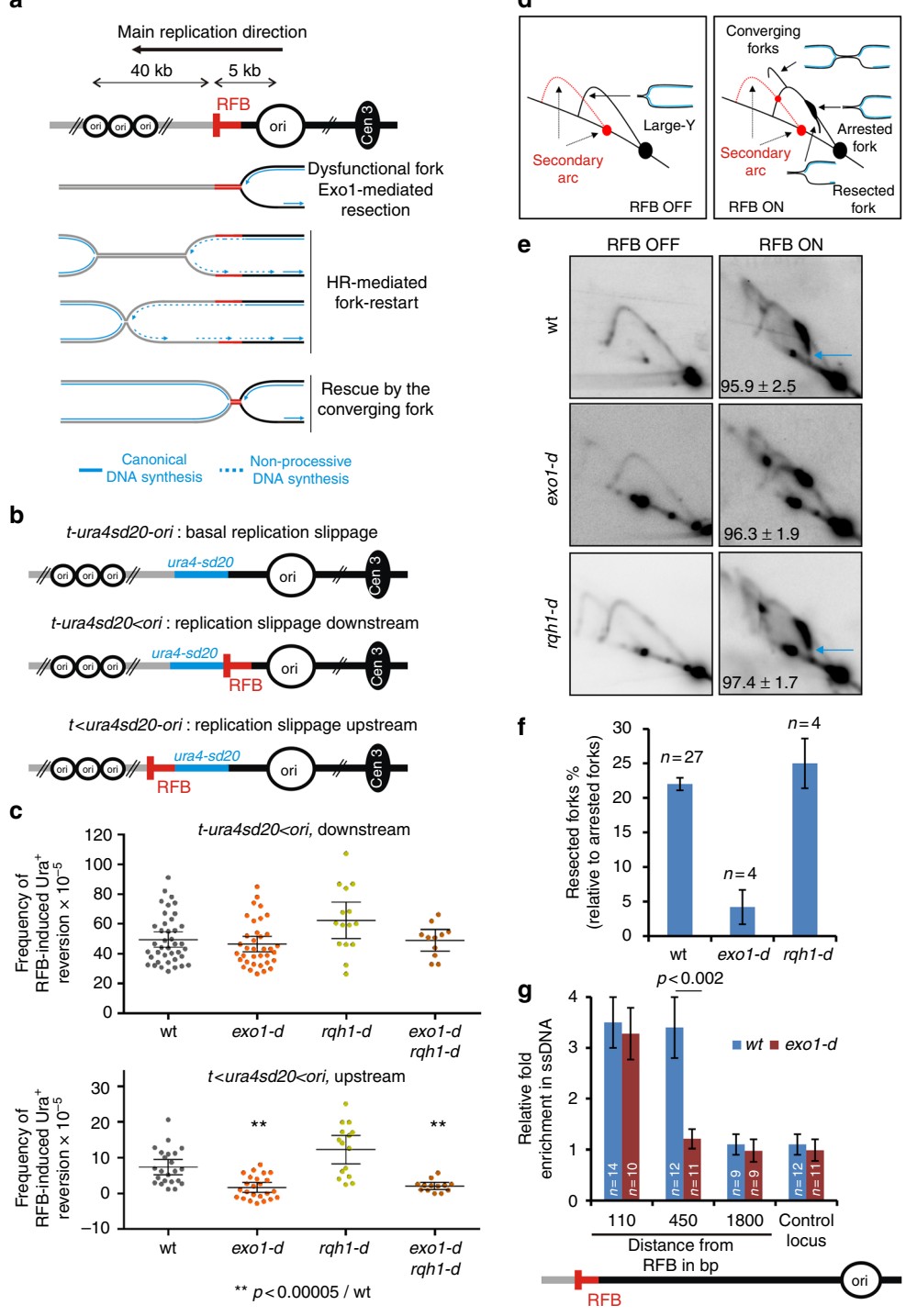

ssDNA gap and not a DSB[17]. Exo1 is the main nuclease responsible for the formation of these gaps, but the lack of Exo1 does not impair the efficiency of replication restart, suggesting that additional nucleases are likely involved[47]. The DSB long-range resection is mediated by two independent parallel pathways: Exo1 and Rqh1[Sgs1/BLM] together with Dna2[27,28]. Upstream RFB-induced Ura$^+$ reversion was abolished in exo1-d cells, whereas downstream RFB-induced Ura$^+$ reversion was not affected (Fig. 1c)[47]. The lack of Rqh1 did not affect upstream and downstream RFB-induced Ura$^+$ reversion, even when exo1 was deleted, indicating that Rqh1 is not required for fork-resection and restart (Fig. 1c).

We recently reported a novel method to monitor fork-resection by bi-dimensional gel electrophoresis (2DGE)[51]. We identified an intermediate originating from the fork arrest signal and descending towards the linear arc, indicative of a loss of mass and shape (Fig. 1d, e, see blue arrow). Alkaline 2DGE showed that this tail signal corresponds to terminally arrested forks containing newly replicated strands undergoing resection[51]. Consistent with this, the tail signal was abolished in exo1-d (Fig. 1e, f) as previously reported[51]. Importantly, the mass of resected forks is consistent with forks arrested at the RTS1-RFB being unbroken. Indeed, a break introduced in one chromatid arm near the fork junction would result in a loss of mass, so intermediates would migrate faster than the monomer. Thus, these analyses of fork-resection by 2DGE further support that end-resection occurs at a DSB-free arrested fork. We analysed fork-resection by 2DGE in rqh1-d cells. The tail signal was unaffected in rqh1-d cells compared to wt cells, confirming that Rqh1 is not part of a long-range resection pathway of terminally arrested forks (Fig. 1e, f).

To get a deeper analysis of ssDNA length generated upstream from the RTS1-RFB, we have recently developed a qPCR assay to directly monitor the presence of ssDNA, based on ssDNA being refractory to restriction digestion[51,52]. As previously reported, in wt cells, ssDNA was enriched 110 bp and 450 bp upstream from the RTS1-RFB but was undetectable at 1.8 kb from the RFB (Fig. 1g). Yet, in the absence of Exo1, ssDNA was still enriched at 110 bp but not at 450 bp. These data hint at the presence of additional nucleases acting on DSB-free arrested forks to generate short ssDNA gap and subsequent replication restart.

**Ctp1 acts with MRN in promoting fork-resection and restart.** We investigated the role of Rad50 and Ctp1 in fork-resection and restart. Compared to wt cells, downstream RFB-induced Ura$^+$ reversion was decreased by 1.8- and 3.4-fold in ctp1-d

and rad50-d cells, respectively (Fig. 2a, upper panel). The ctp1-d rad50-d double mutant showed a defect similar to each single mutant, showing that MRN and Ctp1 act in the same pathway. We estimated that the lack of MRN and Ctp1 results in ~70 % of forks irreversibly terminally arrested at the RTS1-RFB.

As observed in exo1-d cells, upstream RFB-induced Ura$^+$ reversion was similarly abolished in ctp1-d, rad50-d and ctp1-d rad50-d double mutant, suggesting that MRN-Ctp1 act in fork-resection to ensure efficient DNA synthesis resumption (Fig. 2a, bottom panel). Consistent with this, analysis of fork-resection by 2DGE showed that the level of resected forks was severely and similarly decreased in both rad50-d and ctp1-d cells (Fig. 2b, c). We concluded that MRN-Ctp1 act together in promoting fork-resection and subsequent fork-restart.

We reported that Rad52 recruitment to the RTS1-RFB relies on Mre11, independently of its nuclease activity. We analysed two mre11 mutated alleles (Mre11-D65N and Mre11-H134S), defective for both the endo- and exo-nuclease activities[53–55]. These mutants showed no defect in RFB-induced Ura$^+$ reversion, upstream and downstream from the RFB (Supplementary Fig. 2a). Cells expressing the Mre11-D65N mutated form showed no defect in the level of resected forks analysed by 2DGE (Supplementary Fig. 2b, c). Thus, MRN-Ctp1-dependent fork-resection and restart requires an intact MRN complex but not the Mre11 nuclease activity.

**A ~110 bp sized ssDNA gap is necessary to restart fork.** In contrast to Exo1-mediated fork-resection, the MRN-Ctp1-dependent resection is critical to ensure an efficient restart of DNA synthesis. We tested if MRN-Ctp1 acts upstream Exo1 in promoting the formation of ssDNA gaps. In the simultaneous absence of Exo1 and either Ctp1 or Rad50, upstream and downstream RFB-induced Ura$^+$ reversion were decreased to the same extent as the single deletion of either ctp1 or rad50 (Fig. 2a). Fork-resection analysis by 2DGE revealed a similar defect in the single ctp1-d mutant and in the double ctp1-d exo1-d mutant (Fig. 2b, c). These data indicate that the role of MRN-Ctp1 is not redundant with Exo1 function, and that MRN-Ctp1 act upstream Exo1. To clarify the role of these factors in ssDNA gaps formation, we monitored the level of ssDNA by qPCR in ctp1-d and ctp1-d exo1-d cells. The enrichment in ssDNA at 110 bp and 450 bp was dependent on Ctp1. Thus, short and large ssDNA gaps are dependent on the MRN-Ctp1 axis (Fig. 2d). Furthermore, Exo1 was no longer required for ssDNA formation at 110 bp in the absence of Ctp1. As shown for DSB resection[27,28], we propose that fork-resection is a two-step process: MRN-Ctp1

**Fig. 1** Short and long-range resection occurs at terminally arrested forks. **a** Scheme of the RTS1-RFB (red bar) integrated 5 kb away from a strong replication origin (ori). The RTS1-RFB has been integrated at the ura4 locus, where most forks travel from the centromere-proximal origin toward the telomere. "Cen3" indicates centromere position. Distances between replication origins and the RTS1-RFB are indicated. 16 h after thiamine removal, forks traveling from the centromere toward the telomere are blocked in a polar manner. **b** Diagrams of constructs containing the reporter gene ura4-sd20, associated or not to the RTS1-RFB (Supplementary Fig. 1). The ura4-sd20 allele contains a 20 nt duplication flanked by 5 bp of micro-homology[49]. When the ura4-sd20 allele is replicated by a restarted fork, the non-processive DNA synthesis undergoes replication slippage resulting in the deletion of the duplication and the restoration of a functional ura4$^+$ gene. **c** Frequency of upstream and downstream RFB-induced Ura$^+$ reversion. Each dot represents one sample from independent biological replicate. Bars indicate mean values ± 95 % confidence interval (CI). Statistics were calculated using Mann-Whitney U test (Supplementary Data 1). **d** Scheme of replication intermediates (RI) analysed by neutral-neutral 2DGE of the AseI restriction fragment. Psoralen crosslinked DNA samples are prone to partial AseI digestion resulting in a secondary arc which is indicated by red dashed lines in RFB OFF and ON conditions. **e** Representative RI analysis by 2DGE in RFB ON and OFF conditions. Blue arrows indicate DSB-free arrested forks containing nascent strands undergoing resection. Numbers indicate % of blocked forks ± standard deviation (SD). **f** Quantification of resected forks (tail signal), relative to the intensity of arrested forks. Values are means of n samples from independent biological replicates ± 95% CI. **g** Relative enrichment of ssDNA formed upstream from the RTS1-RFB (see Methods). Data show the fold enrichment in ssDNA in the RFB ON relative to OFF condition. A locus on chromosome II is used as control. Values are means of n samples from independent biological replicates ± standard error of the mean (SEM). Statistics were calculated using Mann-Whitney U test

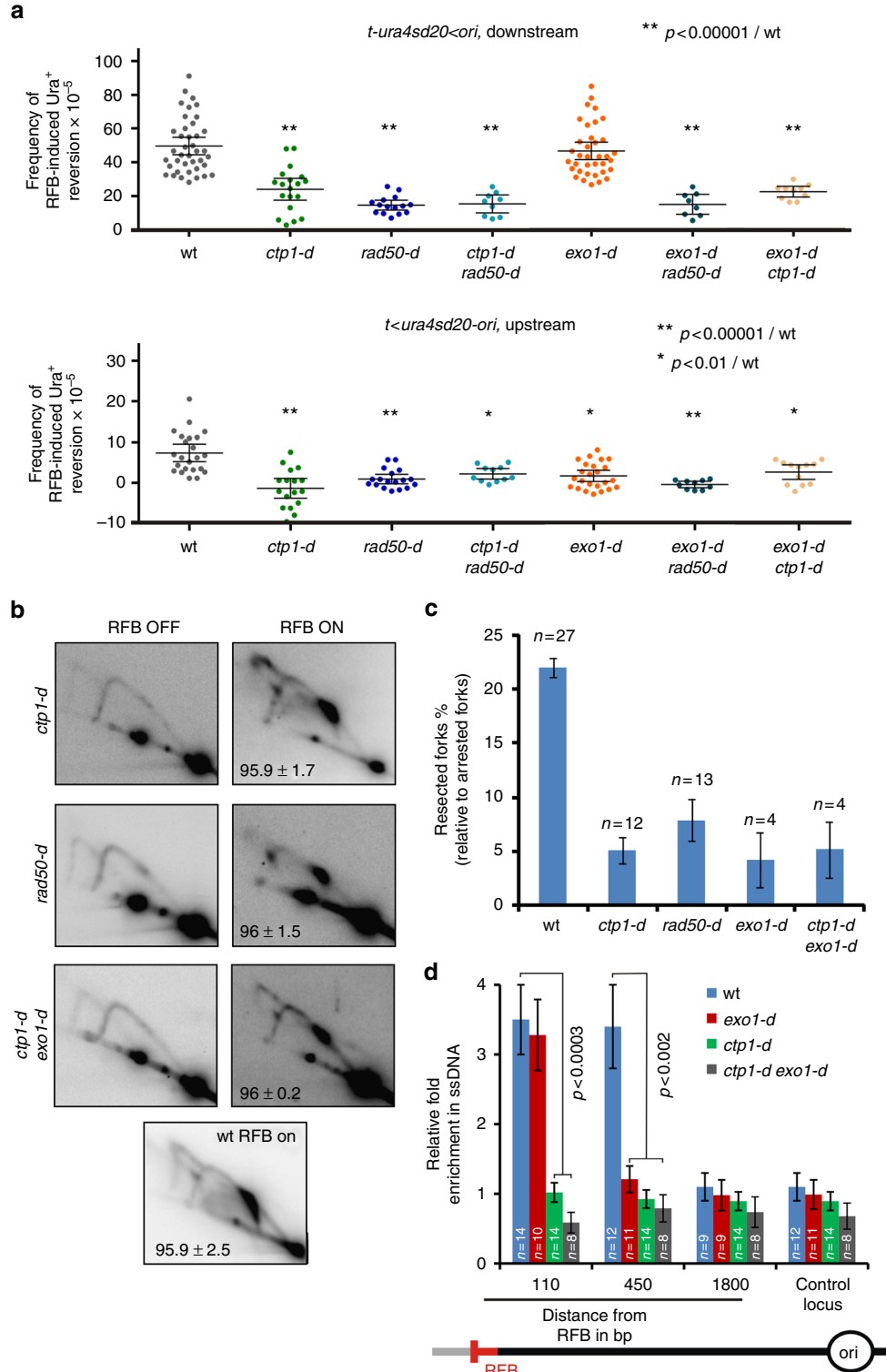

**Fig. 2** MRN-Ctp1-mediated initial fork-resection primes long-range resection by Exo1. **a** Frequency of downstream RFB-induced Ura[+] reversion (top panel, *t-ura4sd20 < ori*) and upstream RFB-induced Ura[+] reversion (bottom panel, *t < ura4sd20-ori*) as described on Fig. 1c. **b** Representative RI analysis by 2DGE in indicated strains upon activation (RFB ON) or not (RFB OFF) of the *RTS1*-RFB, as described on Fig. 1e. **c** Quantification of forks undergoing resection ("tail signal"), relative to the intensity of terminally arrested forks, as described on Fig. 1f. Values are means of *n* samples from independent biological replicates ± 95% CI. **d** Relative enrichment of ssDNA formed upstream from the *RTS1*-RFB, as described on Fig. 1g. Values are means of *n* samples from independent biological replicates ± SEM. Statistics were calculated using the non-parametric Mann-Whitney *U* test. See Supplementary Fig. 2 for the analysis of the *mre11* nuclease dead mutant

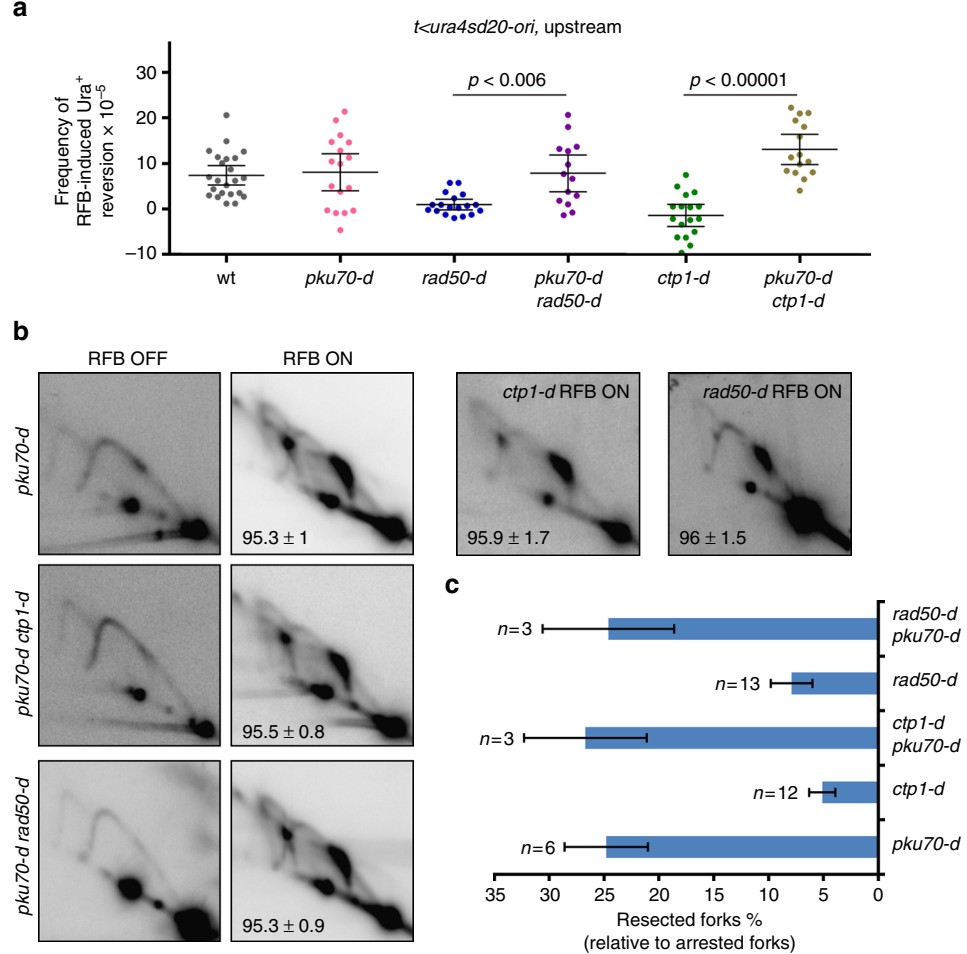

**Fig. 3** The lack of Ku bypasses the initial resection of DSB-free arrested forks. **a** Frequency of upstream RFB-induced Ura+ reversion, as described on Fig. 1c. **b** Representative RI analysis by 2DGE upon activation (RFB ON) or not (RFB OFF) of the *RTS1*-RFB, as described on Fig. 1e. Numbers indicate the % of forks blocked at the *RTS1*-RFB ± SD. **c** Quantification of forks undergoing resection ("tail signal"), relative to the intensity of terminally arrested forks, as described on Fig. 1f. Values are means of *n* samples from independent biological replicates ± 95% CI

promotes the formation of short ssDNA gaps of ~ 110 bp in size which primes Exo1-mediated long-range resection; the initial resection being critical for restarting replication forks, but not the extensive one.

**MRN-Ctp1 removes Ku from terminally arrested forks**. Despite the absence of detectable DSB at the *RTS1*-RFB, we tested the role of Ku in the resection of terminally arrested forks. The deletion of *pku70* on its own did not impact upstream RFB-induced Ura+ reversion, but rescued the defect observed in *rad50-d* and *ctp1-d* cells (Fig. 3a). Analyses by 2DGE allow us to assess the level of resection at the DSB-free arrested fork. We found that the level of resected forks was fully restored in *ctp1-d pku70-d* and *rad50-d pku70-d* cells (Fig. 3b, c). Thus, the lack of Ku bypasses the requirement of MRN-Ctp1 in promoting initial resection of DSB-free arrested forks.

We investigated the recruitment of Pku70 to the *RTS1*-RFB by ChIP-qPCR. In wt cells, Pku70 was recruited to telomeres but no recruitment to the active *RTS1*-RFB was detectable (Fig. 4a, b). Possibly, the binding of Ku to terminally arrested forks is too transitory, as observed for DSBs[36]. In the absence of Rad50, Pku70 accumulated upstream from the active *RTS1*-RFB. A similar recruitment, although to a lesser extent, was observed in *ctp1-d* cells (Fig. 4a). Consistent with the nuclease activity of Mre11 being dispensable in promoting fork-resection, Pku70 was

not detected at the *RTS1*-RFB in *mre11-D65N* cells whereas it was recruited to telomeres (Supplementary Fig. 2d, e). Collectively, these data suggest a role of MRN-Ctp1 in releasing Ku from terminally arrested forks.

The lack of Ku suppresses the sensitivity of *ctp1-d* and *rad50-d* cells not only to DSB-inducing agents but also to replication-blocking agents[19,22,23,36]. These data were interpreted as a role of Ku in binding replication-born DSBs formed in the vicinity of stressed forks. As reported, we found that, in the strains used in this study, the deletion of *pku70* partially rescued the sensitivity of *ctp1-d* and *rad50-d* cells to very low doses of camptothecin (CPT) and methyl methane-sulfonate (MMS) (Supplementary Fig. 3). CPT is an inhibitor of topoisomerase I, which slows down replication fork progression while MMS is an alkylating agent leading to damaged replication forks. Both drugs do not induce detectable DSBs at very low doses[56,57]. We tested whether similar genetic interactions were observed upon induction of the *RTS1*-RFB. Both *ctp1-d* and *rad50-d* cells showed a loss of viability upon induction of the *RTS1*-RFB, that is rescued by deleting *pku70* (Fig. 4c). Altogether, these data support that the inability of MRN-Ctp1 to remove Ku from terminally arrested forks is a lethal event.

**Ku ensures that DSB-free forks are resected in two steps**. In the absence of Ku, MRN-Ctp1 is no longer needed to promote

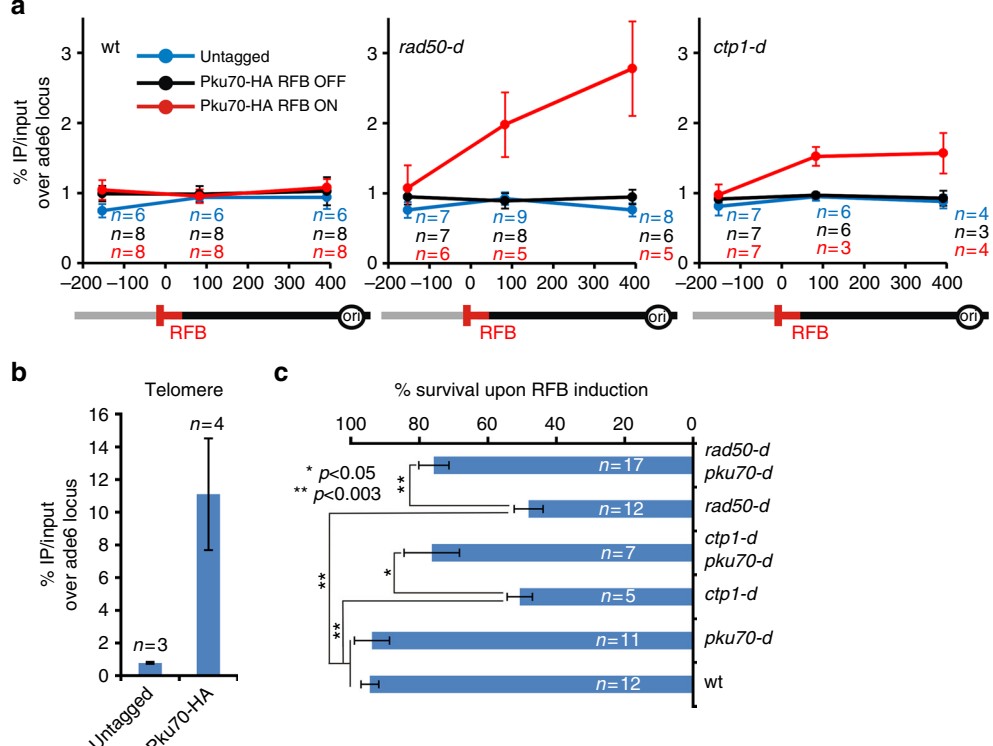

**Fig. 4** MRN-Ctp1 allow Ku eviction from terminally arrested forks. **a** Analysis of Ku recruitment to the *RTS1*-RFB by ChIP-qPCR in indicated strains and conditions. Upstream and downstream distances from the RFB are indicated in base pairs (bp). Values are means of *n* samples from independent biological replicates ± SEM. Statistical analysis was performed using the non-parametric Mann-Whitney *U* test. **b** Analysis of Ku recruitment to telomeres in wt strain. Values are means of *n* samples from independent biological replicates ± SD. **c** Survival of the indicated strains upon activation of the barrier relative to OFF condition. Values are means of *n* samples from independent biological replicates ± SEM. Statistics were calculated using the non-parametric Mann-Whitney *U* test. See Supplementary Fig. 3 for the analysis of cell sensitivity to CPT and MMS

ssDNA gap formation. We tested whether fork-resection is dependent on Exo1 in the absence of Ku. First, upstream RFB-induced Ura$^+$ reversion observed in the *pku70-d ctp1-d* double mutant was dependent on Exo1 (Fig. 5a). Second, 2DGE analysis revealed that fork-resection occurring in *ctp1-d pku70-d* cells was dependent on Exo1 (Fig. 5b, c). Third, as reported by Langerak et al.[36], the suppressive effect of *pku70* deletion on *cpt1-d* cells sensitivity to CPT and MMS was dependent on Exo1 as the *pku70-d cpt1-d exo1-d* triple mutant exhibited the same sensitivity as the single *ctp1-d* mutant (Supplementary Fig. 3). These data establish that the bypass of initial fork-resection by the lack of Ku relies on Exo1. We propose that Ku has an inhibitory effect on the Exo1-mediated long-range fork-resection, requiring a MRN-Ctp1 relief, as proposed for DSBs and telomeres resection.

In the absence of Ku, DSB-free arrested forks are no longer resected in a two-step manner and ssDNA gaps are directly formed by Exo1. Quantification of ssDNA by qPCR showed that ssDNA was significantly more abundant at 1.8 kb upstream from the *RTS1*-RFB in *pku70-d* cells compared to wt cells (Fig. 5d). 2DGE analysis revealed that this extensive fork-resection relies on Exo1 (Fig. 5b, c). Thus, in the absence of Ku, DSB-free arrested forks are excessively resected trough the Exo1-mediated long-range resection. We propose that Ku regulates the formation of ssDNA gaps by ensuring that fork-resection occurs in a two-step manner.

**The lack of Ku delays fork-restart independently of NHEJ.** To test whether the lack of Ku impairs HR-mediated fork-restart, we

applied the downstream RS assay to *pku70-d* cells. RFB-induced Ura$^+$ reversion was decreased by 2.2-fold compared to wt, indicating that the downstream *ura4-sd20* allele is less frequently replicated by a restarted fork in the absence of Ku (Fig. 6a). This cannot be explained by a lower expression of Rad51, Rad52 and RPA (Supplementary Fig. 4a, b). We tested a possible involvement of Ligase 4, and found no defect in upstream and downstream RFB-induced Ura$^+$ reversion in *lig4-d* cells, showing that Ligase 4 is not required to promote fork-resection and subsequent restart (Supplementary Fig. 4c).

We asked whether excessive fork-resection occurring in *pku70-d* cells impacts downstream RFB-induced Ura$^+$ reversion. We analysed the *pku70-d exo1-d* double mutant in which long-range resection is abolished, and the *pku70-d ctp1-d*, and the *pku70-d rad50-d* double mutants, in which long-range resection still occurs (Fig. 5b). The three strains behave as the single *pku70-d* strain, with a ~2-fold reduction in RFB-induced Ura$^+$ reversion compared to wt (Fig. 6a and Supplementary Fig. 4d). Thus, the lack of Ku decreases the frequency at which the downstream *ura4-sd20* allele is replicated by a restarted fork, regardless of the extent of fork-resection and the length of ssDNA gaps.

Next, we tested a strain in which fork convergence from the distal side is minimized by the presence of 10 repeats of the TER2 and TER3 rDNA rRFBs (Fig. 6b). Unlike the *RTS1*-RFB, TER2 and TER3 slow down fork progression without inducing terminally arrested forks and recruitment of HR factors[46]. Delaying the arrival of opposite forks allows more time for the process of HR-mediated fork-restart to occur at the *RTS1*-RFB[50]. We monitored the level of downstream RFB-induced Ura$^+$ reversion in wt and *pku70-d* and observed no significant

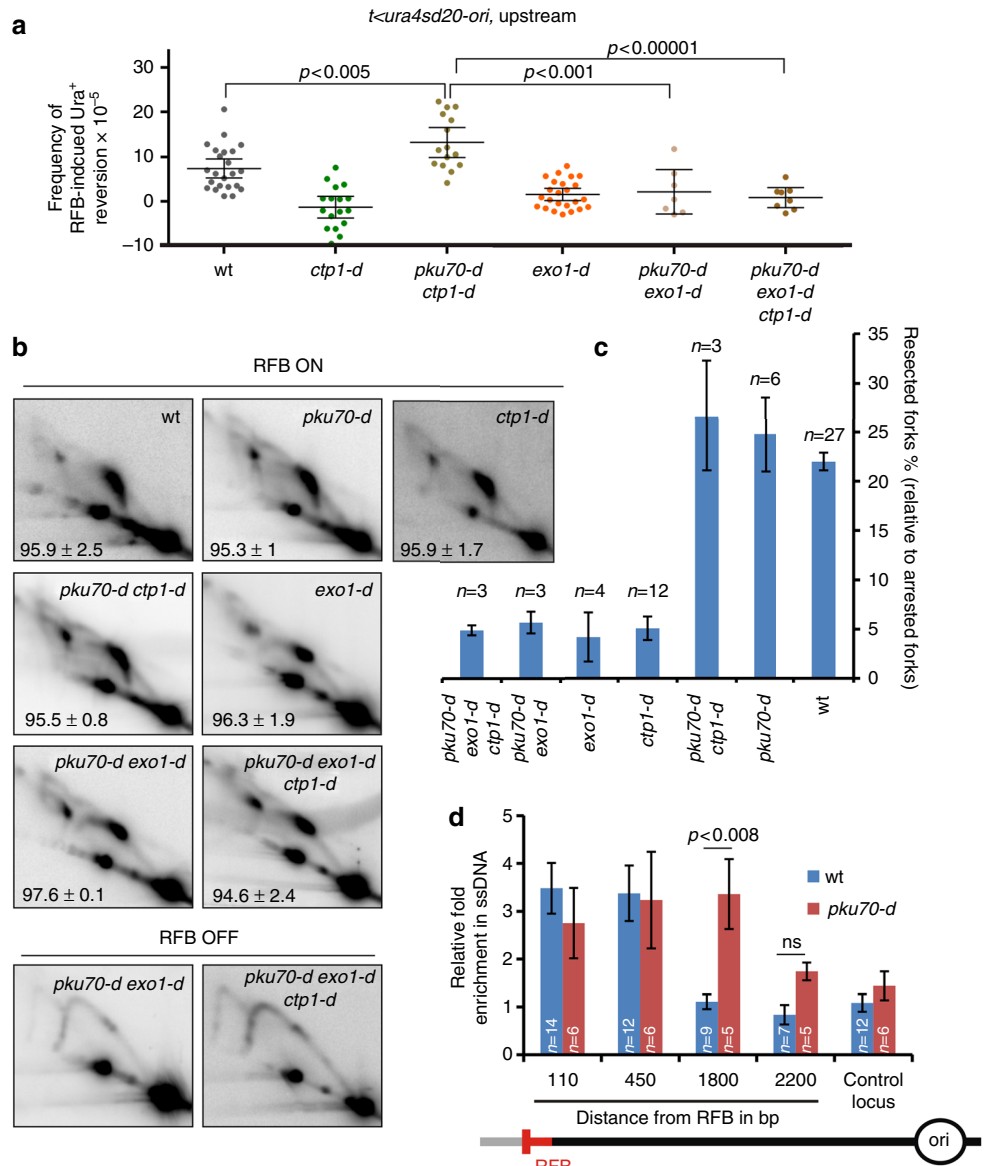

**Fig. 5** Ku ensures that fork-resection occurs as a two-step process. **a** Frequency of upstream RFB-induced Ura$^+$ reversion, as described on Fig. 1c.
**b** Representative RI analysis by 2DGE upon activation (RFB ON) or not (RFB OFF) of the *RTS1*-RFB, as described on Fig. 1e. **c** Quantification of forks undergoing resection ("tail signal"), relative to the intensity of terminally arrested forks, as described on Fig. 1f. Values are means of *n* samples from independent biological replicates ± 95% CI. **d** Relative enrichment of ssDNA formed upstream from the *RTS1*-RFB in indicated strains. The data shown are the fold enrichment in ssDNA in the ON condition (RFB active) relative to the OFF condition (RFB inactive). Values are means of *n* samples from independent biological replicates ± SEM. Statistics were calculated using Mann-Whitney *U* test

differences, even when the Exo1-mediated long-range resection was abolished (Fig. 6b). Thus, the defect in RFB-induced Ura$^+$ reversion is rescued by delaying the arrival of converging forks. These data suggest that the *ura4-sd20* allele is less frequently replicated by a restarted fork in the absence of Ku because of a slow HR-mediated fork-restart process rather than an inability to restart replication forks.

**The lack of Ku impairs RPA and Rad51 recruitment.** We investigated the dynamics of RPA recruitment to the *RTS1*-RFB by fluorescence-based imaging in living cells. We employed a strain in which a GFP-LacI-bound *LacO* array was integrated close to the *RTS1*-RFB, and expressing the RPA subunit Ssb3 fused to mCherry[51] (Fig. 7a). In both wt and *pku70-d* strains, we observed a similar increase in S-phase cells showing RPA

recruitment to the *LacO*-marked RFB (Fig. 7b). Such recruitment did not occur in G2 cells. Thus, RPA is recruited to resected forks in S-phase cells and then evicted once arrested forks have been restarted or rescued by converging forks. Thus, despite that the absence of Ku slows down the process of fork-restart, this does not result in an accumulation of unresolved arrested forks in G2 cells, showing that dysfunctional forks are ultimately rescued by the progression of opposite forks.

When performing cell imaging, we noticed that RPA foci were less bright in the absence of Ku, so we quantified the area and intensity of RPA foci recruited to the *LacO*-marked RFB. We found that the area was not affected, whereas the intensity of RPA foci was decreased by half in *pku70-d* cells compared to wt (Fig. 7c). We have analysed the recruitment of RPA and Rad51 to the *RTS1*-RFB by ChIP-qPCR and found that both RPA and Rad51 were less recruited to arrested forks in *pku70-d* cells

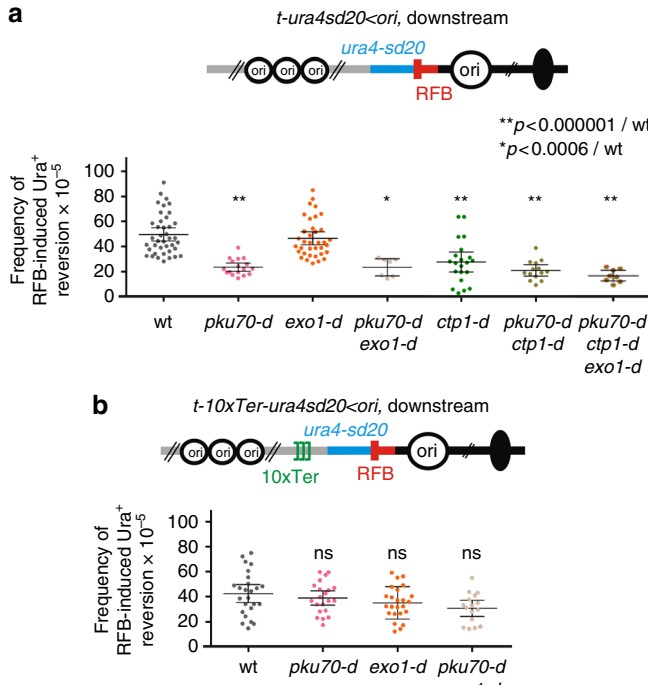

**Fig. 6** The lack of Ku slows down HR-mediated fork-restart. **a** Frequency of downstream RFB-induced Ura$^+$ reversion, as described on Fig. 1c. **b** Frequency of downstream RFB-induced Ura$^+$ reversion using the *t-10xTer-ura4sd20 < ori* construct at which the progression of the converging fork is delayed. Each dot represents one sample from independent biological replicates. Bars indicate mean values $\pm$ 95% CI. Statistics were calculated using the Mann-Whitney *U* test. See Supplementary Fig. 4 for the expression of recombination factors and analysis of the *lig4-d* mutant

compared to wt (Fig. 7d). This defect in recruiting single stranded DNA binding proteins cannot be explained by a reduced amount of ssDNA in the absence of Ku, as ssDNA gaps are ~ twice longer than in wt cells (Fig. 5d). In addition, as observed in budding yeast, we found that Pku70 interacts with RPA, independently of DNA and RNA[58]. Indeed, Pku70-HA co-immuno-precipitated with two RPA subunits, Rad11-YFP and Ssb3-YFP, from protein extracts treated with benzonase (Fig. 7e and Supplementary Fig. 5). Thus, we identify unexpected outcomes at a DSB-free arrested fork in the mere absence of Ku, including extensive fork-resection, reduced recruitment of HR factors and a slow-down on the HR-mediated fork-restart process.

## Discussion

Rad51-dependent processing of replication forks can occur independently of DSB[14,17]. In this work, we investigated the resection step of DSB-free arrested forks, a critical step to expose ssDNA and subsequent recruitment of RPA and Rad51 to promote fork-restart[47]. We made unexpected findings. First, the two-step model of DSB resection applies to DSB-free arrested forks. Secondly, the initial step of fork-resection includes Ku eviction, suggesting that dysfunctional forks undergo fork reversal, providing a single dsDNA end for Ku binding. Third, the lack of Ku impacts several steps of HR-mediated fork processing, independently of NHEJ, resulting in an extensive fork-resection, a reduced recruitment of RPA and Rad51 and a slower HR-mediated fork-restart process.

We show that Ku is recruited to terminally arrested forks. Our previous studies have revealed that no site-specific DSB could be detected at the *RTS1*-RFB, even in the absence of HR[16,17].

Recently, we established that in the absence of Rad52 or Rad51, arrested forks are unprotected and converted into mitotic sister chromatid bridges which favor chromosome breakage randomly during mitosis but not in S-phase[51]. The analysis of fork-derived intermediates by 2DGE relies on the identification of replication intermediates based on their mass and shape. Our 2DGE analyses are consistent with end-resection of nascent strands being initiated from a DSB-free arrested fork[51]. Exploiting the *RTS1*-RFB, our data strongly suggest that Ku is recruited to DSB-free arrested forks at which it ensures that fork-resection occurs as a two-step process. Indeed, we establish that Ku inhibits Exo1-mediated long-range resection of DSB-free arrested forks. MRN-Ctp1 counteracts this inhibition. Ku binds dsDNA ends with high affinity, and with poor affinity for ssDNA[59]. Despite the lack of DSBs, we demonstrate that Ku is recruited to the RFB. These surprising findings favor a model in which DSB-free arrested forks undergo fork reversal[56] (Fig. 8). Reversed forks are DNA structures in which newly replicated strands are annealed together, exposing a single dsDNA end for Ku binding. Fork reversal was shown, in mammals, to occur in response to various replication stresses, such as low doses of CPT and MMS, even in absence of replication-born DSBs[14].

Ku is involved in the repair of replication-born DSB, either generated by defect in processing Okazaki fragments, CPT-induced, or genetically engineered[19,22–24,36]. The mammalian Ku associates with nascent strands after CPT treatment[60] and fission yeast Ku is recruited to the rDNA RFB to stabilize blocked forks[26]. While these data are consistent with Ku being recruited to one ended DSB formed in the vicinity of replication forks, our finding suggest an alternative route to Ku recruitment, independently of DSBs. We propose that the DNA end of reversed fork is recognized and processed as a DSB, despite the absence of detectable DNA breakage (Fig. 8).

MRN and Ctp1 act together to expose ssDNA at DSB-free arrested forks. The lack of the MRN-Ctp1 axis results in Ku persisting at replication forks. Possibly, MRN-Ctp1 initiates fork-resection to create a substrate less favorable to Ku binding[19]. However, the lack of Ku impacts replication and recombination outcomes at the RFB, suggesting that Ku binds arrested forks even if the MRN-Ctp1 axis is functional. Thus, we propose that Ku is recruited early, likely on the reversed arm, and is then quickly evicted by MRN-Ctp1. In support of this, a rapid interplay, occurring within seconds after DNA damage, have been reported between mammalian Ku and MRN in mammals[61].

MRN has a catalytic and structural role in DSBs resection[62]. As observed for the resection of "clean" DSBs[32,36], the Mre11 nuclease activity is dispensable to the process of fork-resection and restart. Langerak et al. have reported that Ku removal from DSBs requires MRN, Ctp1 and the nuclease activity of Mre11[36]. Thus, the requirement of the Mre11 nuclease activity in Ku removal is different at DSB-free arrested forks and DSBs. The tetrameric form of Ctp1 has a scaffolding role in DSBs resection, through DNA binding and bridging activities[63]. Purified Ctp1 shows a binding preference for branched structure containing dsDNA and ssDNA, but no apparent nucleolytic activity was detected in vitro, in contrast to Sae2 and CtIP[63]. We propose that MRN-Ctp1 functions in initiating fork-resection, promoting Ku eviction and HR-mediated fork-restart involve a structural rather than a nucleolytic role. Possibly, MRN-Ctp1 may recruit additional nucleases which remain to be identified.

MRN-Ctp1 and their homologues counteract the accumulation of Ku at one-ended DSB to promote Exo1-dependent resection[23,36,38]. The persistence of Ku on DNA end impairs the recruitment of yeast RPA and mammalian Rad51 without affecting end-resection[24,36]. In fission yeast, a single broken fork is lethal in *ctp1-d* cells[36]. Here, we establish that a single

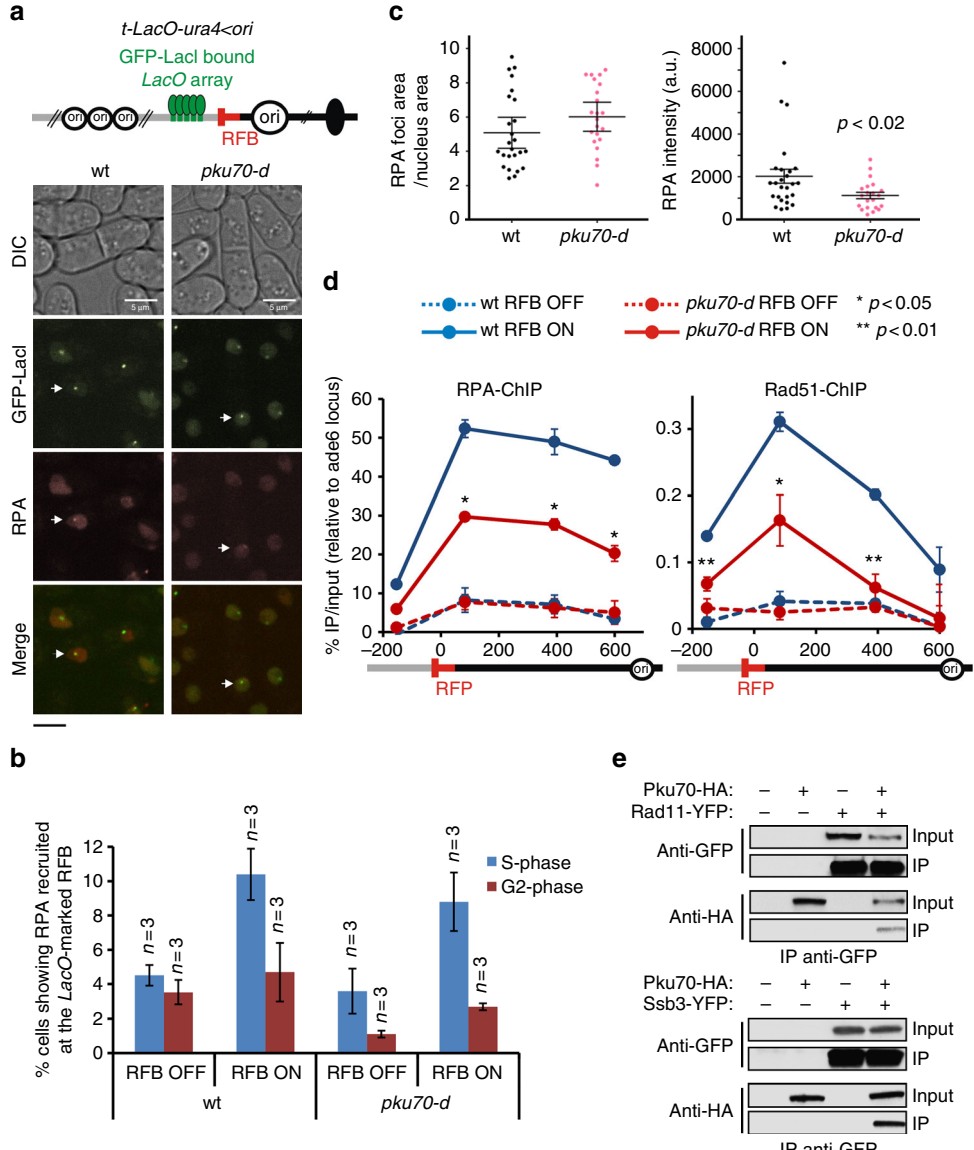

**Fig. 7** The lack of Ku impairs RPA and Rad51 recruitment to terminally arrested forks. **a** Top panel: scheme of the *LacO*-marked *RTS1*-RFB. Bottom panel: representative images showing RPA (labelled with Ssb3-mCherry) recruitment to the *LacO*-marked RFB. Of note, to avoid biases toward a random localization of GFP-LacI and RPA foci, cells with ≥ 3 RPA foci were excluded from the analysis. Scale bars correspond to 5 µm. **b** % of cells showing RPA recruitment to the *LacO*-marked RFB, according to cell cycle phase. G2 and S-phase cells are mononucleated cells and bi-nucleated cells with a septum, respectively. Values are means of *n* samples from independent biological replicates ± SD. **c** Area and intensity of RPA foci recruited to the *LacO*-marked RFB. Each dot represents one RPA foci in S-phase upon RFB activation. Bars indicate the mean value ± 95% CI. Statistics were calculated using the non-parametric Mann-Whitney *U* test. **d** Analysis of RPA and Rad51 recruitment to the *RTS1*-RFB by ChIP-qPCR in indicated strains and conditions. Upstream and downstream distances from the RFB are indicated in base pairs (bp). The values are the means of three to four samples from independent biological replicates ± SEM. Statistical analysis was performed using the Mann-Whitney *U* test. **e** Co-immunoprecipitation of Pku70-HA with two subunits of RPA, Rad11 and Ssb3, fused to the YFP epitope. Experiments were performed in the presence of benzonase. For a longer exposure, see Supplementary Fig. 5

terminally arrested fork is lethal in the absence of either MRN or Ctp1. This lethality is in part caused by the binding of Ku to dysfunctional forks. Thus, we propose that Ku eviction from DSB-free arrested forks by MRN-Ctp1 is an essential step for subsequent replication restart and cell viability.

Together with our previous work, our data establish that the resection of DSB-free arrested forks occurs as a two-step process which is regulated by Ku. The initial MRN-Ctp1-dependent resection promotes Ku eviction and generates ~110 bp of ssDNA, essential to the recruitment of HR factors and subsequent fork-restart[47]. The second step is a long-range resection, mediated by

Exo1, but not Rqh1, generating up to 1 kb of ssDNA which is not strictly required to the resumption of DNA synthesis. Thus, a limited amount of ssDNA is sufficient to promote Rad51-mediated fork-restart, whereas long-range resection may reinforce checkpoint activation.

In the absence of Ku, MRN-Ctp1 is no longer required to initiate fork-resection, which then relies only on Exo1. As a consequence, DSB-free arrested forks are extensively resected with an accumulation of ~2 kb of ssDNA. This supports the notion that Exo1 inhibition by Ku takes place at the initial resection step even with a functional MRN-Ctp1 axis.

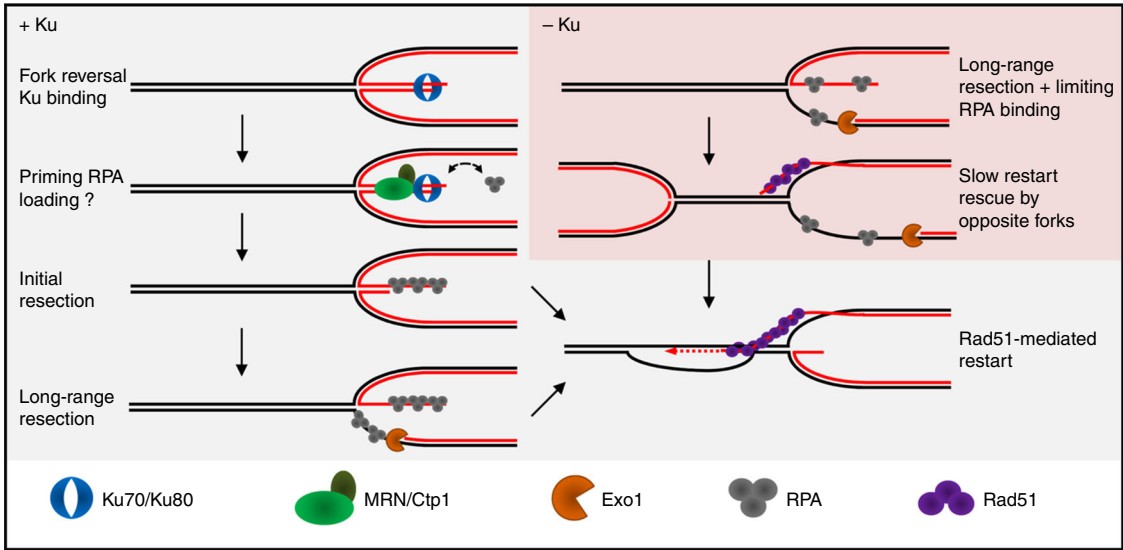

**Fig. 8** Model of two-step resection of DSB-free arrested forks regulated by Ku. In wt cells, DSB-free arrested forks undergo fork reversal providing a single DNA end for Ku binding. MRN-Ctp1 allows Ku removal, generating a 110 bp sized ssDNA on which Rad51 nucleates to promote strand invasion within the restored parental duplex. The initial fork-resection primes the Exo1-mediated long-range resection. Through direct or indirect physical interactions, Ku may be recruited in association with RPA to facilitate its loading on ssDNA. In the absence of Ku, the reversed arm is directly resected by Exo1 resulting in an extensive fork-resection. Possibly, this may destabilize the reversed fork and impact the recruitment of HR factors

Remarkably, the initial resection of DSBs and telomeres is increased in the absence of Ku[21,38]. Thus, Ku plays an important role in ensuring that fork-resection occurs in two steps to avoid unnecessary extensive resection, which can be detrimental to genome stability.

Previous works have proposed a NHEJ-independent role of Ku at replication-born DSBs, to channel repair towards the HR pathway[22,23,36]. Fission yeast genetics indicate a role for Ku in the recovery from replication stress and stabilizing replication forks[25,26]. An important finding we made is that the mere lack of Ku slows down the process of HR-mediated fork-restart, accompanied with a reduced recruitment of RPA and Rad51. These data contrast with Ku acting as a barrier against RPA loading onto ssDNA during DSB repair[36], suggesting replicative-specific functions for Ku. Possibly, Ku may help to transiently stabilize the reversed arm to maintain the fork in a recombination-dependent state. RPA is a critical factor to control end-resection, stability of ssDNA and subsequent recruitment of HR factors[64]. As observed in budding yeast[58], we report physical interactions between Ku and RPA in the absence of exogenous DNA damage, indicating that a cellular fraction of Ku proteins is associated with RPA, directly or indirectly. Thus, Ku may also be recruited to arrested forks in association with RPA to facilitate RPA loading onto ssDNA. We propose that Ku fine-tunes HR-mediated fork-restart by an unknown mechanism (Fig. 8).

Ku allows the recruitment of downstream NHEJ factors such as Ligase 4 to promote DSBs ligation. We found no role for Ligase 4 in promoting fork-resection and restart. Rather, our data suggest that NHEJ inhibition enhances HR-mediated replication restart. Given the potential deleterious outcomes of error-prone NHEJ events on genome stability, it is possible that the recruitment of additional NHEJ factors is prevented to avoid unwanted ligation of the reversed arm at terminally arrested forks.

Recent reports indicate that ssDNA stabilization by RPA influences repair pathway choice at DSB between BIR and microhomology-mediated end joining (MMEJ)[65,66]. MMEJ is an alternative NHEJ mechanism, independent of Ku and Ligase 4 that promotes DSB repair toward chromosomal rearrangement. We propose that the lack of Ku, resulting in reduced RPA and Rad51 recruitment to dysfunctional forks, may favor MMEJ events.

## Methods

**Standard yeast genetics**. Yeast strains used in this work are listed in Supplementary Table 1. Gene deletion and gene tagging were performed by classical and molecular genetics techniques[67]. Strains containing the replication *RTS1*-RFB were grown in supplemented EMM-glutamate media containing thiamine at 60 µM. To induce the *RTS1*-RFB, cells were washed twice in water and grown in supplemented EMM-glutamate media containing thiamine (Rtf1 repressed, RFB OFF condition) or not (Rtf1 expressed, RFB ON condition) for 24 h.

**Analysis of replication intermediates by 2DGE**. Replication intermediates were analyzed by 2DGE as follows[51]. Exponentially growing cells ($2.5 \times 10^9$) were harvested with 0.1% sodium azide and frozen EDTA (80 mM final concentration). Genomic DNA was crosslinked by adding trimethyl psoralen (0.01 mg/ml, TMP, Sigma, 3902–71–4) to the cell suspensions, for 5 min in the dark. Then, cells were exposed to UV-A (365 nm) for 90 s at a flow of 50 mW/cm$^2$. Cells were lysed with 0.625 mg/ml lysing enzyme (Sigma, L1412) and 0.5 mg/ml zymolyase 100 T (Amsbio, 120493-1). The resulting spheroplasts were then embedded into 1% low melting agarose (InCert Agarose, Lonza) plugs, and then incubated overnight at 55 °C in a digestion buffer containing 1 mg/ml of proteinase K (Euromedex EU0090) and then washed and stored in TE (50 mM Tris, 10 mM EDTA) at 4 °C. DNA digestion was performed with 60 units per plug of the restriction enzyme *Ase*I and equilibrated at 0.3 M NaCl. Replication intermediates were enriched using BND cellulose columns (Sigma, B6385) as described in Lambert et al.[17], RIs were migrated in 0.35% agarose gel in TBE for the first dimension. The second dimension was migrated in 0.9% agarose gel-TBE supplemented with EtBr[68]. DNA was transferred to a nylon membrane in 10× SSC. Membranes were incubated with a $^{32}$P radio-labeled *ura4* probe, and RIs were detected using phosphor-imager software (Typhoon-trio) and quantified with ImageQuantTL.

**Live cell imaging**. Cell preparation was done as follows[51]. Cells were grown in supplemented filtered EMM-glutamate, washed twice and resuspended in fresh filtered media without supplements. A drop of 1–2 µl of exponentially growing cultures was placed on a well of a microscope slide (Thermo Scientific) coated with 1.4% ultrapure agarose (Invitrogen, 16500–500) prepared in filtered EMM. Acquisition was performed using an automated spinning disc confocal microscope (Nikon Ti Eclipse inverted) equipped with an ORCA Flash 4.0 camera, a temperature control box set at 30 °C and a 100 × oil immersion objective with a numerical aperture of 1.49. Z-stack image acquisition was performed using the METAMORPH software, and the analysis of the resulting images was performed using ImageJ (https://imagej.nih.gov/ij/). Image acquisition and analysis were performed on workstations of the PICT-IBiSA Orsay Imaging facility of Institut Curie. Ssb3-mCherry and GFP-LacI foci merging/touching was analyzed taking into account cell cycle phase and RFB activation (OFF and ON). "Foci merging/

touching" refers to foci that partially or completely overlap, or to joint foci. The area of Ssb3-mCherry foci that merge with GFP-LacI foci in S-phase cells in ON condition was normalized by the total area of the nucleus. Ssb3-mCherry foci intensity was measured using ImageJ (IntDen); to that value, the nucleus background fluorescence intensity was subtracted.

**DNA extraction and quantification of ssDNA by qPCR**. DNA extraction was performed as follows[51]. 1 to $2 \times 10^8$ cells were mechanically broken by vortexing with glass beads (425–600 μm, Sigma). Genomic DNA was extracted by classical phenol/chloroform extraction. 5 μg of DNA were digested (or not) with 100 units of the restriction enzyme *MseI*. 30 ng of digested and mock-digested DNA were amplified by qPCR using primers surrounding *MseI* restriction site (primers are listed in Supplementary table 2). ssDNA quantification was based on the work of Zierhut & Diffley[52], using the formula:

$$100/\left(\left(1 + [Etarget]^{(\Delta Ct(uncut-cut))}/[Econtrol]^{\Delta Ct(uncut-cut)}\right)/2\right),$$

in which ΔCt is the difference between the threshold cycles of digested (cut) and undigested (uncut) DNA. E is amplification efficiency. A control locus (II-150) that does not contain *MseI* restriction sites was used as a DNA loading control.

**Chromatin immunoprecipitation of Pku70-HA**. Pku70-HA enrichment at *RTS1*-RFB was performed using strains expressing Pku70–3xHA. ChIP experiments were performed as follows[36]. Samples were crosslinked with fresh 1% formaldehyde (Sigma, F-8775) for 15 min. Sonication was performed using a Diagenod Bioruptor at high setting for 8 cycles: 30 s ON + 30 s OFF. Immunoprecipitation was performed using anti-HA antibody coupled to magnetic beads (Thermo Scientific, Pierce 88837, 50 μl in 600 μl final volume). Cell lysis and immunoprecipitation steps were performed in presence of RNase inhibitors (10 units/sample of RNasin, Promega ®). Crosslink was reversed over night at 70 °C. Samples were then incubated with Proteinase K and the DNA was purified using Qiagen PCR purification kit and eluted in 200 μl of water. The relative amount of DNA was quantified by qPCR (primers are listed in Supplementary Table 2). Pku70-HA enrichment was normalized to an internal control locus (*ade6*).

**Chromatin immunoprecipitation of RPA (Ssb3-YFP) and Rad51**. RPA enrichment at *RTS1*-RFB was performed using strains expressing a tagged RPA subunit, Ssb3-YFP. RPA and Rad51 ChIP experiments were performed as follows[47]. Samples were crosslinked with 10 mM DMA (dimethyl adipimidate, Thermos Scientific, 20660) and 1% formaldehyde (Sigma, F-8775). Chromatin was sonicated with a Diagenod Bioruptor set on high for 10 cycles of 30 s ON + 30 s OFF. Immunoprecipitation was performed with an anti-GFP antibody against Ssb3-YFP (Molecular probe, A11122) or anti-Rad51 antibody (Thermo Scientific PA1–4968) at 1:300 and Protein G Dynabeads (Invitrogen, 10003D). The immune-precipitated DNA was purified with a Qiaquick PCR purification kit (Qiagen, 28104) and eluted in 200 μl of water. The relative amount of DNA was determined by qPCR (iQ SYBR green supermix, Biorad, 1708882, primers listed in Table 2). RPA and Rad51 enrichment was normalized relative to an internal control locus (*ade6*).

**Replication slippage assay with *ura4-sd20* allele**. Replication slippage using the *ura4-sd20* allele was performed as follows[49]. 5-FOA resistant colonies were grown on uracil-containing plates with or without thiamine for 2 days at 30 °C, and then inoculated in uracil-containing EMM for 24 h. Cells were diluted and plated on YE plates (for survival counting) and on uracil-free plates containing thiamine to determine the reversion frequency. Colonies were counted after 5 to 7 days of incubation at 30 °C. Statistics were performed using the non-parametric Mann-Whitney U test. Strains used in this study are liable to suppressor accumulation. To avoid taking into account events that do not represent the behavior of each mutant (such as suppressor or other additional spontaneous mutations), outliers were not included in the statistical analysis or graphical representation. Outliers were calculated according to the formula: superior outlier > 1.5 × (Q3-Q1) + Q3, and inferior outlier < 1.5 × (Q3-Q1)-Q1, where Q1 is the first quartile, and Q3 is the third quartile.

**Cell viability**. Cell viability assays were performed as follows[17]. Cells were grown on supplemented EMM without thiamine for 14 h, then they were appropriately diluted and plated on EMM plates with (RFB OFF) or without thiamine (RFB ON). Colonies were counted after 5–7 days incubation at 30 °C and viability was calculated as the ratio of colonies growing in ON condition relative to those growing in OFF condition.

**Co-immunoprecipitation**. $5.10^8$ cells were harvested, washed in cold water and resuspended in 400 μl of EB buffer (50 mM HEPES High salt, 50 mM KOAc pH 7.5, 5 mM EGTA, 1% triton X-100, 1 mM PMSF, and protease inhibitors). Cell lysis was performed with a Precellys homogenizer. The lysate was treated with 250 mU/μl of benzonase for 30 min. After centrifugation, the supernatant

was recovered and an aliquot of 50 μl was saved as the INPUT control. 2 μl of anti-GFP antibody (A11122 from Life Technologies, dilution 1:150) were added to 300 μl of protein extract and incubated for 1 h 30 min at 4 °C on a wheel. Then, 20 μl of Protein G Dynabeads (Invitrogen, 10003D), prewashed in PBS, were added and then incubated at 4 °C overnight. Beads were then washed twice 10 min in EB buffer before migration on acrylamide gel for analysis by Western blot. Pku70-HA, and Ssb3-YFP and Rad11-YFP were detected using anti-HA high-affinity antibody (Roche, 11867423001, 1:500) and using anti-GFP antibody (Roche, 11814460001, 1:1000), respectively. The Supplementary Fig. 5 shows that Pku70-HA was slightly interacting in an unspecific way with anti-GFP antibody. However, the intensity of Pku70-HA in the IP fraction was highly increased in cells expressing SSb3-YFP or Rad11-YFP, showing that most interactions with Pku70-HA are specific.

**Quantification and statistical analysis**. Quantitative densitometric analysis of the Southern-blots (2DGE) was performed using ImageQuant software. The "tail signal" was normalized to the total signal of arrested forks. Cell images were collected using METAMORPH software and analyzed using ImageJ software. The definitions of values and errors bars are mentioned in the figures legend. For most experiments, the number of sample is $n > 3$ obtained from independent biological replicates. Sample size was chosen to demonstrate biological reproducibility and statistical significance if applicable. Statistical analysis was performed using the Mann-Whitney U tests and the Student t test. When no statistics are mentioned, errors bars correspond to the 99 or 95% confidence interval.

**Data availability**. The authors declare that all relevant data supporting the findings of this study are available within the article and its Supplementary Information files or from the corresponding author upon request. Raw data for images and blots have been deposited to Mendeley data and are available at https://data.mendeley.com/datasets/7fpxmtzzxp/1

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

## Acknowledgements

We thank Fuyuki Ishikawa for the gift of the Pku70-HA strain, Tony Carr for the *lig4-d* strain, and Paul Russell for the *mre11-H134S* strain. We thank Vincent Geli for critical comments of the manuscript. We also thank the PICT-IBiSA@Orsay Imaging Facility of the Institut Curie. We thank Soraya Benazzouk and Yasmina Chekkal for their technical assistance. This study was supported by grants from the Institut Curie, the CNRS, the Fondation ARC, the Fondation Ligue (comité Essone), l'Agence Nationale de la Recherche ANR-14-CE10–0010–01, the Institut National du Cancer INCA 2013–1-PLBIO-14, and the Fondation pour la Recherche Médicale "Equipe FRM DEQ20160334889". A.T.-S., A.A.S., and I.I. were funded by the Institut Curie

international PhD program, a French governmental fellowship and the Fondation pour la Recherche Medicale (ING20111223450), respectively. ATS was supported by a 4th year PhD fellowship from Fondation pour la Recherche Medicale (FDT20160435131). The funders had no role in study design, data collection and analysis, the decision to publish, or preparation of the manuscript.

## Author contributions

A.T.-S., A.A.S., I.I., M.C.N., K.F., J.H., and S.A.E.L., performed the experiments. A.T.-S., A.A.S., I.I., and S.A.E.L. designed experiments and analyzed the data. A.T.-S. and S.A.E.L. wrote the paper.

## Additional information

**Competing interests:** The authors declare no competing financial interests.

