## [Peer Review File · Nature Communications]

Reviewers' Comments:

Reviewer #1:

Remarks to the Author:

The authors of this manuscript study the role of enzymes processing 5' strands in replication fork restart. They use fission yeast as model organism and an established assay to follow replication fork stalling at replication fork barrier locus (RFB). To follow resection they used 2D gel analysis and qPCR. However not very quantitative these methods provide general view on the contribution of different enzymes to resection of stalled forks.

Based on the data it is clear that resection at stalled forks shares some features of resection of DSB ends. Namely MRN complex initiates resection and Exo1 is a major or even the only resection enzyme promoting extensive degradation of 5' strand. Second, Ku complex interferes Exo1 mediated resection and this barrier is removed by MRN complex with Ctp1. Thus the mechanism and resection enzymes that work at DNA end at reversed fork appears to be the same as at any other DNA end. The fact that they previously demonstrated the major role of Exo1 in resection at RFB decreased the value of this work. Ku binding at the DNA end of reversed fork and its impact on resection is interesting observation. The manuscript is well written however the role of Ku is overstated and its not clear which data/assay are new here.

Concerns:

Analysis of stalled forks at RFB by 2D gels seems to be inconsistent or not well introduced. With RFB "on" additional structure appears as spot on large Y arc and is very different in wt and mutants but not discussed. What is it and why the intensity is so different? A signal corresponding to 1n is very weak in some gels such as Fig 2b ctp1-d rfb"on", are exposures comparable in all experiments, would longer exposure of this gel make it different? It seems that signal corresponding to resected forks is entirely gone in exo1 but in ctp1 and mre11 mutant some minor smear indicates partial resection. That would be consistent with MRN independent resection that occurs efficiently but with delayed kinetics at DSB ends. What would happen in analysis was done at later time points?

In general MRN complex plays multiple roles in DSB repair, and some of these functions do not relate to resection. Lack of repair in mre11 is interpreted here solely in context of its role in resection. Is it possible that other functions of MRN complex contribute to decreased repair?

It is not clear what is already published and known and what is truly new here. On many occasions one would need to dive into previously published manuscripts to know what is really new result here, or what is a modification of an old system or entirely new assay. In example line 154 says:

"We developed a reporter assay consisting of an inactivated allele... "

Considering that this reporter was published several times it would be better to rephrase this sentence. Which part of Supp Figure 1 is already published?

Another example, the role of Exo1 in resection at RFB was already established by the authors (lack of tail by 2DGE in exo1, Tsang et al.), later published in recent Ait-Saada et al. manuscript (2017). Therefore the title of the chapter should not state that Exo1 is involved in long-range resection (line 182). The news here is that long-range resection is dispensable for replication restart (analysis of rqh1) but even this is not worth a separate chapter as no resection is observed in exo1 single mutant as the authors published previously. In general it is known that Rqh1 has minimal role in resection in fission yeast (Russell's lab work).

In general truly new and non-incremental part of the work starts at line 243, however subtitle "Ku orchestrates initial and long range resection" is an overstatement. Ku has regulatory function in resection, but ku mutant has no deficiency in fork repair and its role is revealed mostly in MRN mutants. The authors show mild phenotypes - fork restart is delayed and RPA loading decreased in ku mutant cells but at the same time ku mutants are not sensitive to DNA damaging agents that

cause many forks stalling such as CPT. Together this work shows that the role of Ku in resection control is similar at DNA ends at reversed forks as at DSB ends or telomeres. The difference is that at DSBs and at telomeres this function is very important in DSB repair pathway choice or telomere protection while at stalled forks there is no consequence in repair.

Reviewer #2:

Remarks to the Author:

Teixeira-Silva present a very interesting study regarding the dynamics of replication fork restart in response to an engineered RTS1 replication fork barrier (RFB). It is a powerful experimental genetic system that has allowed precise examination of the outcomes following replication fork stalling and subsequent restart. This paper centers of the NHEJ-independent role of Ku at stalled forks, which might be paradoxical if forks were not broken such that no double-strand breaks (DSBs) were present. A model is elaborated, supported by strong and substantial experimental evidence, that Ku acts at the single-ended DSB resulting at a reversed fork, a.k.a. "chicken foot".

There are several novel and important findings in the manuscript that significantly enhance our understanding of replication stalling and restart. As such, this study will be of substantial interest to readers interested in replication, DNA repair and recombination, and genome stability. The most notable novel findings are a replication-dependent phenotype of Ku mutants in otherwise wild-type strains; while it is known that Ku can modify the phenotype of other mutants, it has not been clear that Ku alone could change replication outcomes. The most compelling data come from the physical analysis of DNA intermediates at the engineered replication fork barrier, including 2D gels and an ssDNA analysis at different distances from the RFB. These are robust data that strongly support the author's contention of a "two-step" model of fork resection that parallels that seen at typical DSBs, and that, similarly, is influenced by the interplay of the Mre11 complex and Ku at DSB ends followed by long-range action of Exo1. Collectively, the study represents an important contribution to our understanding of replication restart control and its parallels to other recombination processes.

There are issues that need clarification or attention.

1) The authors contention and model are heavily based on the assertion throughout that the RTS1 model does not entail DSB formation at replication forks. However, there is no clear description of the evidence or rationale for this conclusion, it is simply asserted repeatedly. I understand that this experimental system has been used for years, and perhaps this point is well established, but the evidence and rationale must be clearly provided in this paper. How have DSBs, even at low level, been ruled out? This would be appropriate information for the Introduction.

This is a critical point, because as noted correctly by the authors, there have been other studies demonstrating that Ku mutation relieves DSB resection inhibition in Mre11 complex mutants, including at replication forks. Thus, the novelty of the study, and much of the interpretation, hinge on the fact that the DSB end bound by Ku is a reversed, not a broken fork.

It is also a critical point because, as noted below, some phenotypes are relatively weak. Thus, it is important to know findings do not result from a minority process of broken forks.

2) The paper has many places that I find to be somewhat over-interpreted or over-stated.

a) First, the title itself is, to this reviewer, hard to defend. I do not think adequate evidence has been provided that "Ku orchestrates replication fork resection". This would imply that major defects would result from Ku deficiency, when in fact other factors have more profound effects. I don't think it would detract from the findings to use a more balanced language. Even after reading this study, I see the Mre11 complex as being the true "conductor" of the resection process.

b) In the *ura3* reversion assay, even the name used by the authors seems to draw a conclusion that need not be forced. I recommend that it be called what it is, reversion, not concluding that it is always “replication slippage”. More importantly, the authors seem to force a relationship between downstream reversion and the fraction of forks that successfully restart. Reversion may indeed (often) result from restart, but it is beyond what can be known from the assay to directly equate reversion frequency with restart efficiency. Why couldn't a mutant affect one but not the other? Indeed, the authors later make a manipulation (Fig. 5, 10x Ter) that changes the reversion frequency while presumably not changing the behavior of the RST1 RFB (a very nice experiment), demonstrating that the two are not necessarily always coupled. In the end, I think these points are mostly semantic, and will not change the impact of the paper, but I think the text must be rewritten in a manner that does not force conclusions by overly strong equation of an experimental observations to an interpreted cause.

c) Fig 3d. I am concerned about the very small (1.4 fold maximum) enrichment of Ku seen near the RFB. That is a difficult fold enrichment to even be confident of in ChIP, although the error bars do support that it significant (it is critical that these replicates are truly independent, starting from different preparations of cells). But even more importantly, the apparent binding of Ku does not seem to be consistent with the overall model and proposed role of Ku. First, it could not even be detected in wild-type strains, when it most definitely can be at a typical DSB with high fold enrichment (although cell cycle stage might influence this, since RFBs are by definition only operational in S phase when resection is rapid). But in *Mre11* complex mutants, where the 2d gels show a very large accumulation of stalled forks in the face of very little resection near the break in the ssDNA assay, there is still very little Ku apparently bound. Like the conclusion that DSBs do not form, the paper hinges considerably on the robustness of this finding, and its quantitative match to the number of expected lesions, so clarification on these points is warranted.

d) Broadly, Figure 5 addressing the impact of Ku mutation on RPA binding is the weakest part of the paper. Based on the results provided, I think the bullet point conclusion on an impact of Ku on RPA is premature. The weak fold change in the microscopic assay isn't convincing. And the IP showing a Ku-RPA assay is not strong support, given in part non-specific signal shown in Supplemental, coupled with the fact that, even with benzonase treatment, it cannot be concluded that the Ku-RPA interaction is direct. I do not say the conclusion is wrong, simply that more support it required to make it.

3) It is unclear what that basis is for excluding “outliers” in the *ura3* reversion data. I understand this for truly extreme cases (e.g. where every cell has reverted), but when effects are small, only a few fold, outlier exclusion might be affecting results in a way not judgeable by the reader.

4) Overall, the paper is well constructed and logical, including easy to follow figures. However, the manuscript text is very rough in many places. Several additional rounds of editing are required to make it more readable, including attention to many small grammatical issues and other points above. If necessary, consultation on English usage should be considered.

5) It is a small point, but in my reading the term RFB is usually reserved for normal genetic elements that impair replication fork progression; I am not clear that it is common to use it as a global term to encompass all forms of replication inhibition, where many authors might use the terms replication inhibition or replication stress.

Reviewer #3:

Remarks to the Author:

By using a genetic system that blocks a replication fork at a specific locus, the authors investigate resection and restart of terminally-arrested replication forks. Similar to what happens at DNA

double-strand break, they find that MRN and Ctp1 initiate resection at arrested replication forks by removing Ku from DNA. This initial resection allows long-range resection by Exo1. Finally, they show that the lack of Ku increases resection but impairs RPA recruitment, suggesting a role for Ku in recruiting RPA to ssDNA.

The experiments are well done and the results are interesting. However there are two important points that need to be clarified/addressed:

1-The authors propose that "an initial resection mediated by MRN/Ctp1 removes Ku from terminally-arrested forks" (see abstract). However, the Mre11 nuclease activity appears to be dispensable, indicating that Ku removal is not due to a nucleolytic cleavage. In budding yeast, the lack of MRX (but not of Sae2 or Mre11 nuclease activity) increases Ku association to DSBs (Shim et al., 2010), suggesting a competition between Ku and MRX for the same DNA ends rather than a direct MRX-mediated eviction of Ku. However, the authors show that also the lack of Cpt1 (Sae2 in *S. cerevisiae*) (which should affect MRN nuclease activity and not MRN integrity) impairs Ku accumulation. These findings leave me confused because if Mre11 nuclease is not involved, the lack of Cpt1 should mimic the result obtained with the Mre11 nuclease defective mutant. This is an important point that needs to be clarified. Is Ku association impaired in *rad50-d*, *ctp1-d* and *mre11-D65N* cells? Is Mre11-D65N nuclease defective in vitro? Can the authors test other *mre11* nuclease defective allele(s)?

2-The authors propose that Ku helps the recruitment of RPA to ssDNA. To my knowledge there are no evidence that the lack of Ku impairs RPA association at DSBs. Rather, *ku-d* cells hyperactivate a Mec1-dependent checkpoint and therefore they should allow RPA loading onto ssDNA. In addition, since Ku binds the double-stranded DNA, it is not clear to me how Ku could help the association of RPA to ssDNA. The authors should use another assay (ChIP?) to measure quantitatively the association of RPA in Ku mutants because the difference in RPA foci formation between *ku-d* and wt is really very subtle.

Point by point response to reviewer's comments

Reviewer #1 (Remarks to the Author):

The authors of this manuscript study the role of enzymes processing 5' strands in replication fork restart. They use fission yeast as model organism and an established assay to follow replication fork stalling at replication fork barrier locus (RFB). To follow resection they used 2D gel analysis and qPCR. However not very quantitative these methods provide general view on the contribution of different enzymes to resection of stalled forks.

Based on the data it is clear that resection at stalled forks shares some features of resection of DSB ends. Namely MRN complex initiates resection and Exo1 is a major or even the only resection enzyme promoting extensive degradation of 5' strand. Second, Ku complex interferes Exo1 mediated resection and this barrier is removed by MRN complex with Ctp1. Thus the mechanism and resection enzymes that work at DNA end at reversed fork appears to be the same as at any other DNA end. The fact that they previously demonstrated the major role of Exo1 in resection at RFB decreased the value of this work. Ku binding at the DNA end of reversed fork and its impact on resection is interesting observation. The manuscript is well written however the role of Ku is overstated and its not clear which data/assay are new here.

We believe that the novelty of our work is to establish that the end-resection of DSB-free arrested forks occurs in a two-step manner as described for the resection of DSBs, including: an initial resection made by MRN and Ctp1 to prime an Exo1-mediated long range resection, Ku recruitment and its removal by MRN-Ctp1. Given that the mechanisms of end-resection are evolutionarily conserved and the high frequency of fork reversal in response to replication stress in mammals (published by M. Lopes's lab in Zellweger et al. J. Cell Biol 2015), we believe our data present a broad interest to the genome stability field. To highlight the novelty of our findings, and tone down our conclusion regarding the role of Ku, we have modified the title of the manuscript (The End-Joining factor Ku acts in the end-resection of double strand break-free arrested replication forks), the abstract, subheadings of the results section and the main text.

Concerns:

Analysis of stalled forks at RFB by 2D gels seems to be inconsistent or not well introduced. With RFB "on" additional structure appears as spot on large Y arc and is very different in wt and mutants but not discussed. What is it and why the intensity is so different? A signal corresponding to 1n is very weak in some gels such as Fig 2b ctp1-d rfb"on", are exposures comparable in all experiments, would longer exposure of this gel make it different?

We apologize to not have been more careful in the description of 2DGE experiments. The additional signals detected in "RFB ON" conditions result from partial restriction digestion due to the fact that DNA samples are crosslinked with TMP + UVA exposure (described in the methods section). TMP crosslinks AT base pairs which are present in the restriction sites of AseI (ATTAAT), the enzyme used to digest DNA. Partial digestion is unavoidable and gives rise to a secondary arc initiating from a second monomer of higher mass in the "RFB OFF" condition. In the "RFB ON" condition, this

secondary arc contains signal of arrested forks as well. We have modified Fig. 1d to help the interpretation of the 2DGE experiments and mentioned in the figure legend that “Psoralen crosslinked DNA samples are prone to partial *AseI* digestion resulting in a secondary arc which is indicated by red dashed lines in RFB OFF and ON conditions”, (Page 22, lines 853-854).

They are two parameters that can vary from one sample to another. First, partial digestion impacts the number of arc detected. Second, replication intermediates are enriched on BND cellulose which the quality is variable from one batch to another, explaining the variation in the intensity of 1n signal (monomer). The intensity of the 1n signal reflects the efficiency with which linear DNA has been eliminated during the first washes of the enrichment procedure on BND cellulose. Therefore, only the intensity of the RFB signal is used as a reference for quantification and exposure. All 2DGE panels are presented with similar RFB signal intensity. We hope that these explanations will convince the reviewer.

It seems that signal corresponding to resected forks is entirely gone in *exo1* but in *ctp1* and *mre11* mutant some minor smear indicates partial resection. That would be consistent with MRN independent resection that occurs efficiently but with delayed kinetics at DSB ends. What would happen in analysis was done at later time points?

We agree that a very faint “tail” signal is present in some 2DGE experiments for *ctp1* and *mre11* strains. However, we haven’t been able to quantify this tail over the background. The qPCR assay to quantify ssDNA at the active RFB did not reveal accumulation of ssDNA in *ctp1* strain. The 2DGE experiments are performed in “steady state” conditions, 24 hours after thiamine removal. Sixteen hours after thiamine removal, Rft1 is fully expressed and the RFB active. Thus, we estimated that 2DGE experiments are performed in strains experiencing the active RFB during 2 to 3 generations (Lambert et al. Mol Cell 2010). We have previously performed 2DGE experiments 24 hours and 48 hours after thiamine removal and observed no differences in replication intermediates signal.

Strain deleted for *mre11* or *ctp1* are sick and quickly accumulate suppressors. Despite having performed experiments using freshly defrost strains, we cannot exclude that the faint tail “signal” reflects a fraction of cells in which suppressor mutations did occur allowing fork-resection.

In general MRN complex plays multiple roles in DSB repair, and some of these functions do not relate to resection. Lack of repair in *mre11* is interpreted here solely in context of its role in resection. Is it possible that other functions of MRN complex contribute to decreased repair?

We agree with this comment. We cannot exclude that the role of Mre11 in promoting fork-resection and fork-restart are separable. However, we can only speculate on this point. The topic of this work was to define the mechanisms of end-resection at a DSB-free arrested fork. We have not investigated potential separation-of-function *mre11* mutants to define if its role in fork-restart is genetically separable from its role in fork-resection. We have modified a paragraph in the discussion to address this point “We propose that MRN-Ctp1 functions in initiating fork-resection, promoting Ku eviction and HR-mediated fork-restart involves a structural rather than a nucleolytic role, (Page 11, lines 390-39).

It is not clear what is already published and known and what is truly new here. On many occasions one would need to dive into previously published manuscripts to know what is really new result here,

or what is a modification of an old system or entirely new assay. In example line 154 says:

“We developed a reporter assay consisting of an inactivated allele... “

Considering that this reporter was published several times it would be better to rephrase this sentence. Which part of Supp Figure 1 is already published?

We apologize for our writing and presentation of the data not spotlighting better our novel findings. As mentioned above, the subheadings of the results section have been modified to integrate the reviewers' comments.

The sentence has been rewritten: “We have previously developed a reporter assay consisting of an inactivated allele of *ura4*, *ura4-sd20*, which allows us to infer the degree of RS caused by the restarted fork by monitoring the frequency of Ura⁺ reversion upon expression of Rtf1”, (page 5, lines 148-11). Also, the first and second paragraphs of the results section have been largely modified. Since the reporter assays have been previously published, the details of the genetics assays have been moved to the legend of Fig. 1 and supplementary Fig. 1.

The Supplementary Fig. 1 A has been published in Iraqui et al. PLoS Genetics 2012. Data of panel B are novel and have not been published.

Another example, the role of Exo1 in resection at RFB was already established by the authors (lack of tail by 2DGE in *exo1*, Tsang et al.), later published in recent Ait-Saada et al. manuscript (2017). Therefore the title of the chapter should not state that Exo1 is involved in long-range resection (line 182). The news here is that long-range resection is dispensable for replication restart (analysis of *rqh1*) but even this is not worth a separate chapter as no resection is observed in *exo1* single mutant as the authors published previously. In general it is known that Rqh1 has minimal role in resection in fission yeast (Russell's lab work).

The subheading has been changed for “Short and long-range resection occurs at terminally-arrested forks”, (Page 5, line 166). We agree that the role of Exo1 in the long-range resection of arrested forks has been previously published: the lack of tail by 2DGE in Ait Saada et al. Mol Cell, 2017 and the RFB-induced RS in Tsang et al. J. of Cell Science 2014. However, our qPCR assay to monitor ssDNA formation at the RFB revealed that fork-resection is not fully abolished in the *exo1* single mutant (Fig. 1g), showing that a short and long-range-resection occurs at arrested forks.

We agree that our data on Rqh1 is not the most innovative part of the manuscript, but we felt it was important to test its role in fork-resection, in combination with *exo1* deletion. Therefore data on the single mutant *exo1-d* are provided alongside the single *rqh1* mutant and the double *exo1 rqh1* mutant. The 2DGE experiments are different blots from those published in Ait Saada et al. Mol Cell 2017, and RFB-induced RS have been performed alongside the *rqh1-d* mutant. Our data confirm Russell's work: Rqh1 has minimal role in the resection at DSBs and replication forks.

In general truly new and non-incremental part of the work starts at line 243, however subtitle “Ku orchestrates initial and long range resection” is an overstatement. Ku has regulatory function in resection, but *ku* mutant has no deficiency in fork repair and its role is revealed mostly in MRN mutants. The authors show mild phenotypes - fork restart is delayed and RPA loading decreased in *ku* mutant cells but at the same time *ku* mutants are not sensitive to DNA damaging agents that cause many forks stalling such as CPT. Together this work shows that the role of Ku in resection control is

similar at DNA ends at reversed forks as at DSB ends or telomeres. The difference is that at DSBs and at telomeres this function is very important in DSB repair pathway choice or telomere protection while at stalled forks there is no consequence in repair.

We believe that describing the role of Ctp1 in promoting fork-resection and Rad51-mediated fork restart is novel (Fig. 2, Ctp1 acts with MRN in promoting fork-resection and restart, (Page 6, line 199). The data establishing that fork-resection is a two-step process (initial resection followed by a long-range resection) (Fig. 2; A ~110 bp sized ssDNA gap is sufficient and necessary to restart fork, (page 7, line 220) are also novel.

Regarding the role of Ku in regulating fork-resection, we have changed the title and extensively modified the main text. We also provide additional data showing that the recruitment of RPA and Rad51 to the active RFB is decreased in the absence of Pku70 (Fig. 7d) (page 9, lines 319-331). Collectively, our data established that the mere lack of Pku70 (when MRN and Ctp1 are functional) impacts several steps of the HR-mediated fork processing, including an extensive resection of the fork, a reduced RPA and Rad51 recruitment and a slower HR-mediated fork restart process (see discussion, page 10, lines 350-352). We agree that these phenotypes are not sufficient to result in cell sensitivity to replication-blocking agents in the absence of Pku70, but these phenotypes may contribute to increase genome instability upon replication stress. Similarly, the single *exo1-d* mutant shows a strong defect in the long-range resection, but no delay in fork restart and no high cellular sensitivity to replication stress. Thus, we believe that defining the role of factors involved in the resection of DSB-free arrested forks is important.

Reviewer #2 (Remarks to the Author):

Teixeira-Silva present a very interesting study regarding the dynamics of replication fork restart in response to an engineered RTS1 replication fork barrier (RFB). It is a powerful experimental genetic system that has allowed precise examination of the outcomes following replication fork stalling and subsequent restart. This paper centers on the NHEJ-independent role of Ku at stalled forks, which might be paradoxical if forks were not broken such that no double-strand breaks (DSBs) were present. A model is elaborated, supported by strong and substantial experimental evidence, that Ku acts at the single-ended DSB resulting at a reversed fork, a.k.a. “chicken foot”.

There are several novel and important findings in the manuscript that significantly enhance our understanding of replication stalling and restart. As such, this study will be of substantial interest to readers interested in replication, DNA repair and recombination, and genome stability. The most notable novel findings are a replication-dependent phenotype of Ku mutants in otherwise wild-type strains; while it is known that Ku can modify the phenotype of other mutants, it has not been clear that Ku alone could change replication outcomes. The most compelling data come from the physical analysis of DNA intermediates at the engineered replication fork barrier, including 2D gels and an ssDNA analysis at different distances from the RFB. These are robust data that strongly support the author’s contention of a “two-step” model of fork resection that parallels that seen at typical DSBs, and that, similarly, is influenced by the interplay of the Mre11 complex and Ku at DSB ends followed by long-range action of Exo1. Collectively, the study represents an important contribution to our understanding of replication restart control and its parallels to other recombination processes.

There are issues that need clarification or attention.

1) The authors contention and model are heavily based on the assertion throughout that the RTS1 model does not entail DSB formation at replication forks. However, there is no clear description of the evidence or rationale for this conclusion, it is simply asserted repeatedly. I understand that this experimental system has been used for years, and perhaps this point is well established, but the evidence and rationale must be clearly provided in this paper. How have DSBs, even at low level, been ruled out? This would be appropriate information for the Introduction.

This is a critical point, because as noted correctly by the authors, there have been other studies demonstrating that Ku mutation relieves DSB resection inhibition in Mre11 complex mutants, including at replication forks. Thus, the novelty of the study, and much of the interpretation, hinge on the fact that the DSB end bound by Ku is a reversed, not a broken fork.

It is also a critical point because, as noted below, some phenotypes are relatively weak. Thus, it is important to know findings do not result from a minority process of broken forks.

We thank the reviewer for this comment on a critical point of our work. Our rationale to exclude the formation of a DSB at the RFB is:

a) No site-specific DSBs were detectable at the active RTS1-RFB by Pulse Field Gel Electrophoresis and Southern-blot, even in the absence of HR (Lambert al. Mol Cell 2010; Mizuno et al. Gene & Dev, 2009).

b) The analysis of recombination intermediates at the *RTS1*-RFB was consistent with fork-restart occurring through a template switch event initiated by an ssDNA gap and not a DSB (Lambert et al. Mol Cell 2010)

c) In the absence of Rad52 or Rad51, arrested forks are unprotected and converted into mitotic sister chromatid bridges which favor chromosome breakage randomly during mitosis and not in S-phase (Ait Saada et al. Mol Cell 2017)

d) Using 2DGE, the mass of resected forks are consistent with forks arrested at the *RTS1*-RFB being unbroken. Indeed, a break introduced in one chromatid arm near the fork junction would result in a loss of mass and intermediates would migrate faster than the monomer. Thus, these analyses of fork-resection by 2DGE further support that end-resection occurs at a DSB-free arrested forks (Ait Saada et al. Mol Cell, 2017 and this present study).

This rationale has been introduced at the appropriate places in the main text (result section: page 5, lines 168-171 and page 6 lines 184-188; discussion section: page 10, lines 353-359). We believe that these modifications strengthen our conclusion that Ku regulates the end-resection of DSB-free arrested forks.

2) The paper has many places that I find to be somewhat over-interpreted or over-stated.

a) First, the title itself is, to this reviewer, hard to defend. I do not think adequate evidence has been provided that “Ku orchestrates replication fork resection”. This would imply that major defects would result from Ku deficiency, when in fact other factors have more profound effects. I don’t think it would detract from the findings to use a more balanced language. Even after reading this study, I see the Mre11 complex as being the true “conductor” of the resection process.

We agree with this comment and the title of the manuscript has been changed for “The End-Joining factor Ku acts in the end-resection of double strand break-free arrested replication forks”.

b) In the *ura3* reversion assay, even the name used by the authors seems to draw a conclusion that need not be forced. I recommend that it be called what it is, reversion, not concluding that it is always “replication slippage”. More importantly, the authors seem to force a relationship between downstream reversion and the fraction of forks that successfully restart. Reversion may indeed (often) result from restart, but it is beyond what can be known from the assay to directly equate reversion frequency with restart efficiency. Why couldn’t a mutant affect one but not the other? Indeed, the authors later make a manipulation (Fig. 5, 10x Ter) that changes the reversion frequency while presumably not changing the behavior of the *RST1* RFB (a very nice experiment), demonstrating that the two are not necessarily always coupled. In the end, I think these points are mostly semantic, and will not change the impact of the paper, but I think the text must be rewritten in a manner that does not force conclusions by overly strong equation of an experimental observation to an interpreted cause.

Our previous works have established that the frequency of replication slippage induced by the RFB reflects the frequency at which the *ura4-sd20* allele is replicated by a restarted fork (Iraqi et al. PLoS Genetics 2012, Tsang et al. J. of Cell Science 2014, Ait Saada et al. Mol Cell, 2017). Nonetheless, we

agree with the reviewer that what is scored is the frequency of Ura⁺ reversion that we used as readout of the frequency at which the *ura4-sd20* allele is replicated by a restarted fork in the cell population. Therefore, the expression “RFB-induced RS” has been change for “RFB-induced Ura⁺ reversion” in the manuscript, including the main and supplementary figures and their respective legends and in the main text. Since the reporter assays have been previously published, the details of the genetics assays have been moved to the legend of Fig 1 and supplementary Fig 1. We added in the main text the following sentence “We have previously developed a reporter assay consisting of an inactivated allele of *ura4*, *ura4-sd20*, which allows us to infer the degree of RS caused by the restarted fork by monitoring the frequency of Ura⁺ reversion upon expression of Rtf1 (Supplementary Fig. 1). The frequency of Ura⁺ reversion is used as readout of the frequency at which the *ura4-sd20* allele is replicated by a restarted fork in the cell population”, (page 5, lines 148-152).

c) Fig 3d. I am concerned about the very small (1.4 fold maximum) enrichment of Ku seen near the RFB. That is a difficult fold enrichment to even be confident of in ChIP, although the error bars do support that it significant (it is critical that these replicates are truly independent, starting from different preparations of cells). But even more importantly, the apparent binding of Ku does not seem to be consistent with the overall model and proposed role of Ku. First, it could not even be detected in wild-type strains, when it most definitely can be at a typical DSB with high fold enrichment (although cell cycle stage might influence this, since RFBs are by definition only operational in S phase when resection is rapid). But in *Mre11* complex mutants, where the 2d gels show a very large accumulation of stalled forks in the face of very little resection near the break in the ssDNA assay, there is still very little Ku apparently bound. Like the conclusion that DSBs do not form, the paper hinges considerably on the robustness of this finding, and its quantitative match to the number of expected lesions, so clarification on these points is warranted.

First, we apologize that the figure was mislabeled. The original figure did not represent the % IP relative to the RFB OFF condition but the fold enrichment over the RFB OFF condition. In the revised version, we provide new ChIP experiments, after optimizing our protocol (changes in the protocol have been introduced in the Methods section, page 14, lines 215-528). We have also introduced an internal control which is the recruitment of Pku70-HA to telomeres (Fig. 4a and b, Supplementary Fig. 2d and e). We have obtained better enrichment in the *rad50-d* mutant (around 2 to 3%). It is important to keep in mind that only few Ku molecules are expected to be recruited to the RFB, in contrast to other proteins such as RPA and Rad51 which form oligomers. We remain unable to observe Pku70 recruitment to the active RFB in wt strain, whereas Pku70 is recruited to telomeres in the same ChIP experiments (Fig. 4a and b). Similar data were obtained for the *mre11-D65N* mutant (Supplementary Fig. 2d and e). Cell synchronization has failed to provide better data. We concluded that the recruitment of Ku to the arrested fork is too transient, being quickly evicted by MRN. In support of this, it has been estimated that mammalian Ku binds DSBs in less than a second and that MRN is recruited to DSBs within 10-30 seconds; this rapid interplay between MRN and Ku occurring within second after DNA damage (Hartlerode et al. NSMB, 2015). This point has been added to the discussion section (page 10, lines 380-382). Finally, we observed that the lack of Ku in a context in which the MRN complex is functional, impacts several step of HR-mediated fork processing, including an extensive fork resection (Fig. 5d), a slower HR-mediated fork restart process (Fig. 6) and a decreased RPA and Rad51 recruitment (Fig. 7a-d). Therefore, we favor a model in which Ku is recruited to the active RFB in the wt cells. This point has been added in the discussion section (page 10, lines 376-382).

We would like to strengthen that 2DGE are performed after enrichment of replication intermediates genomic DNA extracted from an asynchronous cell population. Therefore, the intensity of the RFB appears as well enriched. Also, as mentioned above, the analysis of fork-resection by 2DGE indicates that Ku regulates the resection of DSB-free arrested forks (page 6, lines 184-188). On the principle, we cannot formally exclude that a minor fraction of the active RFB experiences a DSB to which Ku is recruited. Collectively, our genetics and molecular data do not support this scenario. However, to remain rigorous in our interpretation, we have stated in the manuscript that Ku is recruited to dysfunctional forks and avoided the statement that Ku is recruited to DSB-free arrested forks (page 7, line 236).

d) Broadly, Figure 5 addressing the impact of Ku mutation on RPA binding is the weakest part of the paper. Based on the results provided, I think the bullet point conclusion on an impact of Ku on RPA is premature. The weak fold change in the microscopic assay isn't convincing. And the IP showing a Ku-RPA assay is not strong support, given in part non-specific signal shown in Supplemental, coupled with the fact that, even with benzonase treatment, it cannot be concluded that the Ku-RPA interaction is direct. I do not say the conclusion is wrong, simply that more support it required to make it.

In the revised version, we provide data showing that RPA and Rad51 recruitment to the RFB are reduced in the *pku70-d* mutant (Fig. 7d) (page 9, lines 322-326), further supporting our microscopy studies. We have down toned our conclusion and stated in the abstract that "The mere lack of Ku impacts the processing of arrested forks, leading to an extensive resection, a reduced recruitment of RPA and Rad51 and a slower fork-restart process." The main text has been rewritten accordingly in the results section. We have added a paragraph in the discussion section to discuss the mechanism by which Ku may fine tune HR-mediated fork restart (page 11, lines 415-427).

3) It is unclear what that basis is for excluding "outliers" in the *ura3* reversion data. I understand this for truly extreme cases (e.g. where every cell has reverted), but when effects are small, only a few fold, outlier exclusion might be affecting results in a way not judgeable by the reader.

The replication slippage (or *Ura*⁺ reversion) assay starts by picking single colonies on 5FOA-containing plates which are then growth for several generations on non-selective media then plated on selective media (to select for *Ura*⁺ reversion) or not (for cell survival). Despite the work being conducted using freshly defrost strains, many strains used in this study are liable to suppressor accumulations (such as *rad50-d*, *ctp1-d* and corresponding double mutants) which are unavoidable. Suppressors may impact replication outcomes at the RFB but also cell survival. Thus, we have decided to exclude outliers from our analysis (explained in the methods section pages 14-15, lines 541-552). Most of scatter plots include from 10 up to 39 samples from independent biological replicates. We are confident that the number of samples analyzed makes our statistical analysis robust enough to support our conclusion.

4) Overall, the paper is well constructed and logical, including easy to follow figures. However, the manuscript text is very rough in many places. Several additional rounds of editing are required to make it more readable, including attention to many small grammatical issues and other points above. If necessary, consultation on English usage should be considered.

Grammatical errors were corrected.

5) It is a small point, but in my reading the term RFB is usually reserved for normal genetic elements that impair replication fork progression; I am not clear that it is common to use it as a global term to encompass all forms of replication inhibition, where many authors might use the terms replication inhibition or replication stress.

The paragraph in the introduction has been rewritten accordingly to this comment (page 3, lines 62-65).

Reviewer #3 (Remarks to the Author):

By using a genetic system that blocks a replication fork at a specific locus, the authors investigate resection and restart of terminally-arrested replication forks. Similar to what happens at DNA double-strand break, they find that MRN and Ctp1 initiate resection at arrested replication forks by removing Ku from DNA. This initial resection allows long-range resection by Exo1. Finally, they show that the lack of Ku increases resection but impairs RPA recruitment, suggesting a role for Ku in recruiting RPA to ssDNA.

The experiments are well done and the results are interesting. However there are two important points that need to be clarified/addressed:

1-The authors propose that “an initial resection mediated by MRN/Ctp1 removes Ku from terminally-arrested forks” (see abstract). However, the Mre11 nuclease activity appears to be dispensable, indicating that Ku removal is not due to a nucleolytic cleavage. In budding yeast, the lack of MRX (but not of Sae2 or Mre11 nuclease activity) increases Ku association to DSBs (Shim et al., 2010), suggesting a competition between Ku and MRX for the same DNA ends rather than a direct MRX-mediated eviction of Ku. However, the authors show that also the lack of Ctp1 (Sae2 in *S. cerevisiae*) (which should affect MRN nuclease activity and not MRN integrity) impairs Ku accumulation. These findings leave me confused because if Mre11 nuclease is not involved, the lack of Ctp1 should mimic the result obtained with the Mre11 nuclease defective mutant. This is an important point that needs to be clarified. Is Ku association impaired in *rad50-d*, *ctp1-d* and *mre11-D65N* cells? Is Mre11-D65N nuclease defective in vitro? Can the authors test other *mre11* nuclease defective allele(s)?

In the revised version, we provide new ChIP experiments, after optimizing our protocol (changes in the protocol have been introduced in the Methods section, page 14, lines 517-528). We have also introduced an internal control which is the recruitment of Pku70-HA to telomeres (Fig. 4a and b, Supplementary Fig. 2d and e). In the absence of Rad50, Pku70 accumulated upstream from the active *RTS1*-RFB. A similar recruitment, although to a lesser extent, was observed in *ctp1-d* cells (Fig. 4a) (page 7, lines 244-251). We have analysed the *mre11-D65N* mutant and found no accumulation of Pku70 at the active RFB whereas it is recruited to telomeres (Supplementary Fig. 2 d-e). We have analysed another nuclease dead *mre11* mutant (*mre11-H134S-Myc*, send by the lab of P. Russell) and found no defect in downstream and upstream RFB-induced replication slippage in this mutant (Supplementary Fig. 2a and b). We haven't tested in vitro the nuclease activity of these mutated forms of Mre11. However, the Mre11-D65N and Mre11-H134S are equivalent to the well-characterized *S. cerevisiae* Mre11-D56N and Mre11-H125N, respectively, which have been shown to have negligible exonuclease activity and a complete loss of endonuclease activity in vitro (Krogh et al. Genetics 2005, Moreau et al. MCB 1999). These data further support that the nuclease activity of Mre11 is dispensable to promote Ku eviction from arrested forks and initiate end-resection. We have proposed in the discussion that the role of MRN and Ctp1 in promoting fork-resection and restart involved a structural role rather than a nucleolytic role (page 11, lines 383-393).

By analyzing the resection of DSBs in fission yeast, Langerak and colleagues have reported that end-resection requires MRN and Ctp1 but not the nuclease activity of Mre11 (using the *mre11-H134S*

mutant). In contrast, Ku removal requires MRN, Ctp1 and the nuclease activity of Mre11. Therefore, the requirement of the Mre11 nuclease activity in Ku removal is different at DSB-free arrested forks and DSBs. We have added this point to the discussion (page 10, lines 383-387).

It has not been shown in vitro that fission yeast Ctp1 stimulates the nuclease activity of Mre11 in vitro. Thus, we have rewritten the sentence “The endo- and exo-nuclease activities of Mre11, stimulated by Sae2/Ctp1, are not strictly required to DNA end resection at “clean” DSBs” in “The endo- and exo-nuclease activities of Mre11, stimulated by Sae2, are not strictly required to DNA end resection at “clean” DSBs” (page 3, lines 92-95).

We agree that the question of competition between Ku and MRN for DSBs binding versus the alternative possibility that Ku binds DSBs and is then evicted by MRN is a longstanding question. We agree that the fact that we observed no Ku recruitment to the RFB in wt cells (Fig. 4a) may favor the second hypothesis. However, we report that the sole lack of Ku impacts replication outcomes at the RFB, including extensive fork resection, a reduced RPA and Rad51 recruitment and a slow HR-mediated fork restart process. The mere lack of Ku (when MRN and Ctp1 are functional) impacts the processing of arrested forks, leading to an extensive resection, a reduced recruitment of RPA and Rad51 and a slower fork-restart process (abstract). Thus, we believe that collectively, our genetics and molecular data favors a model in which Ku is recruited to arrested forks and then evicted by MRN and Ctp1. We added in the discussion section: “Possibly, MRN-Ctp1 initiates fork-resection to create a substrate less favorable to Ku binding¹⁹. However, the lack of Ku impacts replication and recombination outcomes at the RFB, suggesting that Ku binds arrested forks even if the MRN-Ctp1 axis is functional. Thus, we propose that Ku is recruited early, likely on the reversed arm, and is then quickly evicted by MRN-Ctp1. In support of this, a rapid interplay, occurring within seconds after DNA Damage, have been reported between mammalian Ku and MRN in mammals” (page 10, lines 377-382).

2-The authors propose that Ku helps the recruitment of RPA to ssDNA. To my knowledge there are no evidence that the lack of Ku impairs RPA association at DSBs. Rather, *ku-d* cells hyperactivate a Mec1-dependent checkpoint and therefore they should allow RPA loading onto ssDNA. In addition, since Ku binds the double-stranded DNA, it is no clear to me how Ku could help the association of RPA to ssDNA. The authors should use another assay (ChIP?) to measure quantitatively the association of RPA in Ku mutants because the difference in RPA foci formation between *ku-d* and wt is really very subtle.

In the revised version, we provide data showing that RPA and Rad51 recruitment to the RFB are reduced in the *pkv70-d* mutant (Fig. 7d) (page 9, lines 322-326), further supporting our microscopy studies. We have down toned our conclusion and state in the abstract that “The mere lack of Ku impacts the processing of arrested forks, leading to an extensive resection, a reduced recruitment of RPA and Rad51 and a slower fork-restart process”. The main text has been rewritten accordingly in the results section. We have added a paragraph in the discussion section to discuss the mechanism by which Ku may fine tune HR-mediated fork restart (page 11, lines 415-427).

Reviewers' Comments:

Reviewer #1:

Remarks to the Author:

The revised manuscript is substantially improved and most of the concerns were properly addressed.

Reviewer #2:

Remarks to the Author:

Teixeira-Silva present a revised version of their study on the role of Ku in replication restart. In the context of all reviewer comments, I still find that the observations of a role of Ku, and a phenotype in its absence, at replication in otherwise normal cells to be a substantially important finding. I find the revised document to be much more balanced with respect to claims and presentation on this point. The paper also provides additional information regarding the resection process happening at stalled/collapsed forks, even if some of these points are sometimes incremental relative to current knowledge. Specific points of interest are the relationship of short- and long-range resection and data supporting the contention that restart is delayed but not necessarily abrogated in the absence of Ku. I will not repeat my prior specific points of concern; I will simply say that I think they have all been satisfactorily addressed in the revision, including improved ChIP studies. In total, I am happy to endorse this interesting study for publication in Nature Communication.

Reviewer #3:

Remarks to the Author:

The revision nicely addressed most of the reviewers' comments. The manuscript is now significantly improved and suitable for publication.

Point by point response to reviewer's comments

Reviewer #1 (Remarks to the Author):

The authors of this manuscript study the role of enzymes processing 5' strands in replication fork restart. They use fission yeast as model organism and an established assay to follow replication fork stalling at replication fork barrier locus (RFB). To follow resection they used 2D gel analysis and qPCR. However not very quantitative these methods provide general view on the contribution of different enzymes to resection of stalled forks.

Based on the data it is clear that resection at stalled forks shares some features of resection of DSB ends. Namely MRN complex initiates resection and Exo1 is a major or even the only resection enzyme promoting extensive degradation of 5' strand. Second, Ku complex interferes Exo1 mediated resection and this barrier is removed by MRN complex with Ctp1. Thus the mechanism and resection enzymes that work at DNA end at reversed fork appears to be the same as at any other DNA end. The fact that they previously demonstrated the major role of Exo1 in resection at RFB decreased the value of this work. Ku binding at the DNA end of reversed fork and its impact on resection is interesting observation. The manuscript is well written however the role of Ku is overstated and its not clear which data/assay are new here.

We believe that the novelty of our work is to establish that the end-resection of DSB-free arrested forks occurs in a two-step manner as described for the resection of DSBs, including: an initial resection made by MRN and Ctp1 to prime an Exo1-mediated long range resection, Ku recruitment and its removal by MRN-Ctp1. Given that the mechanisms of end-resection are evolutionarily conserved and the high frequency of fork reversal in response to replication stress in mammals (published by M. Lopes's lab in Zellweger et al. J. Cell Biol 2015), we believe our data present a broad interest to the genome stability field. To highlight the novelty of our findings, and tone down our conclusion regarding the role of Ku, we have modified the title of the manuscript (The End-Joining factor Ku acts in the end-resection of double strand break-free arrested replication forks), the abstract, subheadings of the results section and the main text.

Concerns:

Analysis of stalled forks at RFB by 2D gels seems to be inconsistent or not well introduced. With RFB "on" additional structure appears as spot on large Y arc and is very different in wt and mutants but not discussed. What is it and why the intensity is so different? A signal corresponding to 1n is very weak in some gels such as Fig 2b ctp1-d rfb"on", are exposures comparable in all experiments, would longer exposure of this gel make it different?

We apologize to not have been more careful in the description of 2DGE experiments. The additional signals detected in "RFB ON" conditions result from partial restriction digestion due to the fact that DNA samples are crosslinked with TMP + UVA exposure (described in the methods section). TMP crosslinks AT base pairs which are present in the restriction sites of AseI (ATTAAT), the enzyme used to digest DNA. Partial digestion is unavoidable and gives rise to a secondary arc initiating from a second monomer of higher mass in the "RFB OFF" condition. In the "RFB ON" condition, this

secondary arc contains signal of arrested forks as well. We have modified Fig. 1d to help the interpretation of the 2DGE experiments and mentioned in the figure legend that “Psoralen crosslinked DNA samples are prone to partial *AseI* digestion resulting in a secondary arc which is indicated by red dashed lines in RFB OFF and ON conditions”, (Page 22, lines 853-854).

They are two parameters that can vary from one sample to another. First, partial digestion impacts the number of arc detected. Second, replication intermediates are enriched on BND cellulose which the quality is variable from one batch to another, explaining the variation in the intensity of 1n signal (monomer). The intensity of the 1n signal reflects the efficiency with which linear DNA has been eliminated during the first washes of the enrichment procedure on BND cellulose. Therefore, only the intensity of the RFB signal is used as a reference for quantification and exposure. All 2DGE panels are presented with similar RFB signal intensity. We hope that these explanations will convince the reviewer.

It seems that signal corresponding to resected forks is entirely gone in *exo1* but in *ctp1* and *mre11* mutant some minor smear indicates partial resection. That would be consistent with MRN independent resection that occurs efficiently but with delayed kinetics at DSB ends. What would happen in analysis was done at later time points?

We agree that a very faint “tail” signal is present in some 2DGE experiments for *ctp1* and *mre11* strains. However, we haven’t been able to quantify this tail over the background. The qPCR assay to quantify ssDNA at the active RFB did not reveal accumulation of ssDNA in *ctp1* strain. The 2DGE experiments are performed in “steady state” conditions, 24 hours after thiamine removal. Sixteen hours after thiamine removal, Rft1 is fully expressed and the RFB active. Thus, we estimated that 2DGE experiments are performed in strains experiencing the active RFB during 2 to 3 generations (Lambert et al. Mol Cell 2010). We have previously performed 2DGE experiments 24 hours and 48 hours after thiamine removal and observed no differences in replication intermediates signal.

Strain deleted for *mre11* or *ctp1* are sick and quickly accumulate suppressors. Despite having performed experiments using freshly defrost strains, we cannot exclude that the faint tail “signal” reflects a fraction of cells in which suppressor mutations did occur allowing fork-resection.

In general MRN complex plays multiple roles in DSB repair, and some of these functions do not relate to resection. Lack of repair in *mre11* is interpreted here solely in context of its role in resection. Is it possible that other functions of MRN complex contribute to decreased repair?

We agree with this comment. We cannot exclude that the role of Mre11 in promoting fork-resection and fork-restart are separable. However, we can only speculate on this point. The topic of this work was to define the mechanisms of end-resection at a DSB-free arrested fork. We have not investigated potential separation-of-function *mre11* mutants to define if its role in fork-restart is genetically separable from its role in fork-resection. We have modified a paragraph in the discussion to address this point “We propose that MRN-Ctp1 functions in initiating fork-resection, promoting Ku eviction and HR-mediated fork-restart involves a structural rather than a nucleolytic role, (Page 11, lines 390-39).

It is not clear what is already published and known and what is truly new here. On many occasions one would need to dive into previously published manuscripts to know what is really new result here,

or what is a modification of an old system or entirely new assay. In example line 154 says:

“We developed a reporter assay consisting of an inactivated allele... “

Considering that this reporter was published several times it would be better to rephrase this sentence. Which part of Supp Figure 1 is already published?

We apologize for our writing and presentation of the data not spotlighting better our novel findings. As mentioned above, the subheadings of the results section have been modified to integrate the reviewers' comments.

The sentence has been rewritten: “We have previously developed a reporter assay consisting of an inactivated allele of *ura4*, *ura4-sd20*, which allows us to infer the degree of RS caused by the restarted fork by monitoring the frequency of Ura⁺ reversion upon expression of Rtf1”, (page 5, lines 148-11). Also, the first and second paragraphs of the results section have been largely modified. Since the reporter assays have been previously published, the details of the genetics assays have been moved to the legend of Fig. 1 and supplementary Fig. 1.

The Supplementary Fig. 1 A has been published in Iraqui et al. PLoS Genetics 2012. Data of panel B are novel and have not been published.

Another example, the role of Exo1 in resection at RFB was already established by the authors (lack of tail by 2DGE in *exo1*, Tsang et al.), later published in recent Ait-Saada et al. manuscript (2017). Therefore the title of the chapter should not state that Exo1 is involved in long-range resection (line 182). The news here is that long-range resection is dispensable for replication restart (analysis of *rqh1*) but even this is not worth a separate chapter as no resection is observed in *exo1* single mutant as the authors published previously. In general it is known that Rqh1 has minimal role in resection in fission yeast (Russell's lab work).

The subheading has been changed for “Short and long-range resection occurs at terminally-arrested forks”, (Page 5, line 166). We agree that the role of Exo1 in the long-range resection of arrested forks has been previously published: the lack of tail by 2DGE in Ait Saada et al. Mol Cell, 2017 and the RFB-induced RS in Tsang et al. J. of Cell Science 2014. However, our qPCR assay to monitor ssDNA formation at the RFB revealed that fork-resection is not fully abolished in the *exo1* single mutant (Fig. 1g), showing that a short and long-range-resection occurs at arrested forks.

We agree that our data on Rqh1 is not the most innovative part of the manuscript, but we felt it was important to test its role in fork-resection, in combination with *exo1* deletion. Therefore data on the single mutant *exo1-d* are provided alongside the single *rqh1* mutant and the double *exo1 rqh1* mutant. The 2DGE experiments are different blots from those published in Ait Saada et al. Mol Cell 2017, and RFB-induced RS have been performed alongside the *rqh1-d* mutant. Our data confirm Russell's work: Rqh1 has minimal role in the resection at DSBs and replication forks.

In general truly new and non-incremental part of the work starts at line 243, however subtitle “Ku orchestrates initial and long range resection” is an overstatement. Ku has regulatory function in resection, but *ku* mutant has no deficiency in fork repair and its role is revealed mostly in MRN mutants. The authors show mild phenotypes - fork restart is delayed and RPA loading decreased in *ku* mutant cells but at the same time *ku* mutants are not sensitive to DNA damaging agents that cause many forks stalling such as CPT. Together this work shows that the role of Ku in resection control is

similar at DNA ends at reversed forks as at DSB ends or telomeres. The difference is that at DSBs and at telomeres this function is very important in DSB repair pathway choice or telomere protection while at stalled forks there is no consequence in repair.

We believe that describing the role of Ctp1 in promoting fork-resection and Rad51-mediated fork restart is novel (Fig. 2, Ctp1 acts with MRN in promoting fork-resection and restart, (Page 6, line 199). The data establishing that fork-resection is a two-step process (initial resection followed by a long-range resection) (Fig. 2; A ~110 bp sized ssDNA gap is sufficient and necessary to restart fork, (page 7, line 220) are also novel.

Regarding the role of Ku in regulating fork-resection, we have changed the title and extensively modified the main text. We also provide additional data showing that the recruitment of RPA and Rad51 to the active RFB is decreased in the absence of Pku70 (Fig. 7d) (page 9, lines 319-331). Collectively, our data established that the mere lack of Pku70 (when MRN and Ctp1 are functional) impacts several steps of the HR-mediated fork processing, including an extensive resection of the fork, a reduced RPA and Rad51 recruitment and a slower HR-mediated fork restart process (see discussion, page 10, lines 350-352). We agree that these phenotypes are not sufficient to result in cell sensitivity to replication-blocking agents in the absence of Pku70, but these phenotypes may contribute to increase genome instability upon replication stress. Similarly, the single *exo1-d* mutant shows a strong defect in the long-range resection, but no delay in fork restart and no high cellular sensitivity to replication stress. Thus, we believe that defining the role of factors involved in the resection of DSB-free arrested forks is important.

Reviewer #2 (Remarks to the Author):

Teixeira-Silva present a very interesting study regarding the dynamics of replication fork restart in response to an engineered RTS1 replication fork barrier (RFB). It is a powerful experimental genetic system that has allowed precise examination of the outcomes following replication fork stalling and subsequent restart. This paper centers on the NHEJ-independent role of Ku at stalled forks, which might be paradoxical if forks were not broken such that no double-strand breaks (DSBs) were present. A model is elaborated, supported by strong and substantial experimental evidence, that Ku acts at the single-ended DSB resulting at a reversed fork, a.k.a. “chicken foot”.

There are several novel and important findings in the manuscript that significantly enhance our understanding of replication stalling and restart. As such, this study will be of substantial interest to readers interested in replication, DNA repair and recombination, and genome stability. The most notable novel findings are a replication-dependent phenotype of Ku mutants in otherwise wild-type strains; while it is known that Ku can modify the phenotype of other mutants, it has not been clear that Ku alone could change replication outcomes. The most compelling data come from the physical analysis of DNA intermediates at the engineered replication fork barrier, including 2D gels and an ssDNA analysis at different distances from the RFB. These are robust data that strongly support the author’s contention of a “two-step” model of fork resection that parallels that seen at typical DSBs, and that, similarly, is influenced by the interplay of the Mre11 complex and Ku at DSB ends followed by long-range action of Exo1. Collectively, the study represents an important contribution to our understanding of replication restart control and its parallels to other recombination processes.

There are issues that need clarification or attention.

1) The authors contention and model are heavily based on the assertion throughout that the RTS1 model does not entail DSB formation at replication forks. However, there is no clear description of the evidence or rationale for this conclusion, it is simply asserted repeatedly. I understand that this experimental system has been used for years, and perhaps this point is well established, but the evidence and rationale must be clearly provided in this paper. How have DSBs, even at low level, been ruled out? This would be appropriate information for the Introduction.

This is a critical point, because as noted correctly by the authors, there have been other studies demonstrating that Ku mutation relieves DSB resection inhibition in Mre11 complex mutants, including at replication forks. Thus, the novelty of the study, and much of the interpretation, hinge on the fact that the DSB end bound by Ku is a reversed, not a broken fork.

It is also a critical point because, as noted below, some phenotypes are relatively weak. Thus, it is important to know findings do not result from a minority process of broken forks.

We thank the reviewer for this comment on a critical point of our work. Our rationale to exclude the formation of a DSB at the RFB is:

a) No site-specific DSBs were detectable at the active RTS1-RFB by Pulse Field Gel Electrophoresis and Southern-blot, even in the absence of HR (Lambert al. Mol Cell 2010; Mizuno et al. Gene & Dev, 2009).

b) The analysis of recombination intermediates at the *RTS1*-RFB was consistent with fork-restart occurring through a template switch event initiated by an ssDNA gap and not a DSB (Lambert et al. Mol Cell 2010)

c) In the absence of Rad52 or Rad51, arrested forks are unprotected and converted into mitotic sister chromatid bridges which favor chromosome breakage randomly during mitosis and not in S-phase (Ait Saada et al. Mol Cell 2017)

d) Using 2DGE, the mass of resected forks are consistent with forks arrested at the *RTS1*-RFB being unbroken. Indeed, a break introduced in one chromatid arm near the fork junction would result in a loss of mass and intermediates would migrate faster than the monomer. Thus, these analyses of fork-resection by 2DGE further support that end-resection occurs at a DSB-free arrested forks (Ait Saada et al. Mol Cell, 2017 and this present study).

This rationale has been introduced at the appropriate places in the main text (result section: page 5, lines 168-171 and page 6 lines 184-188; discussion section: page 10, lines 353-359). We believe that these modifications strengthen our conclusion that Ku regulates the end-resection of DSB-free arrested forks.

2) The paper has many places that I find to be somewhat over-interpreted or over-stated.

a) First, the title itself is, to this reviewer, hard to defend. I do not think adequate evidence has been provided that “Ku orchestrates replication fork resection”. This would imply that major defects would result from Ku deficiency, when in fact other factors have more profound effects. I don’t think it would detract from the findings to use a more balanced language. Even after reading this study, I see the Mre11 complex as being the true “conductor” of the resection process.

We agree with this comment and the title of the manuscript has been changed for “The End-Joining factor Ku acts in the end-resection of double strand break-free arrested replication forks”.

b) In the *ura3* reversion assay, even the name used by the authors seems to draw a conclusion that need not be forced. I recommend that it be called what it is, reversion, not concluding that it is always “replication slippage”. More importantly, the authors seem to force a relationship between downstream reversion and the fraction of forks that successfully restart. Reversion may indeed (often) result from restart, but it is beyond what can be known from the assay to directly equate reversion frequency with restart efficiency. Why couldn’t a mutant affect one but not the other? Indeed, the authors later make a manipulation (Fig. 5, 10x Ter) that changes the reversion frequency while presumably not changing the behavior of the *RST1* RFB (a very nice experiment), demonstrating that the two are not necessarily always coupled. In the end, I think these points are mostly semantic, and will not change the impact of the paper, but I think the text must be rewritten in a manner that does not force conclusions by overly strong equation of an experimental observation to an interpreted cause.

Our previous works have established that the frequency of replication slippage induced by the RFB reflects the frequency at which the *ura4-sd20* allele is replicated by a restarted fork (Iraqi et al. PLoS Genetics 2012, Tsang et al. J. of Cell Science 2014, Ait Saada et al. Mol Cell, 2017). Nonetheless, we

agree with the reviewer that what is scored is the frequency of Ura⁺ reversion that we used as readout of the frequency at which the *ura4-sd20* allele is replicated by a restarted fork in the cell population. Therefore, the expression “RFB-induced RS” has been change for “RFB-induced Ura⁺ reversion” in the manuscript, including the main and supplementary figures and their respective legends and in the main text. Since the reporter assays have been previously published, the details of the genetics assays have been moved to the legend of Fig 1 and supplementary Fig 1. We added in the main text the following sentence “We have previously developed a reporter assay consisting of an inactivated allele of *ura4*, *ura4-sd20*, which allows us to infer the degree of RS caused by the restarted fork by monitoring the frequency of Ura⁺ reversion upon expression of Rtf1 (Supplementary Fig. 1). The frequency of Ura⁺ reversion is used as readout of the frequency at which the *ura4-sd20* allele is replicated by a restarted fork in the cell population”, (page 5, lines 148-152).

c) Fig 3d. I am concerned about the very small (1.4 fold maximum) enrichment of Ku seen near the RFB. That is a difficult fold enrichment to even be confident of in ChIP, although the error bars do support that it is significant (it is critical that these replicates are truly independent, starting from different preparations of cells). But even more importantly, the apparent binding of Ku does not seem to be consistent with the overall model and proposed role of Ku. First, it could not even be detected in wild-type strains, when it most definitely can be at a typical DSB with high fold enrichment (although cell cycle stage might influence this, since RFBs are by definition only operational in S phase when resection is rapid). But in *Mre11* complex mutants, where the 2d gels show a very large accumulation of stalled forks in the face of very little resection near the break in the ssDNA assay, there is still very little Ku apparently bound. Like the conclusion that DSBs do not form, the paper hinges considerably on the robustness of this finding, and its quantitative match to the number of expected lesions, so clarification on these points is warranted.

First, we apologize that the figure was mislabeled. The original figure did not represent the % IP relative to the RFB OFF condition but the fold enrichment over the RFB OFF condition. In the revised version, we provide new ChIP experiments, after optimizing our protocol (changes in the protocol have been introduced in the Methods section, page 14, lines 215-528). We have also introduced an internal control which is the recruitment of Pku70-HA to telomeres (Fig. 4a and b, Supplementary Fig. 2d and e). We have obtained better enrichment in the *rad50-d* mutant (around 2 to 3%). It is important to keep in mind that only few Ku molecules are expected to be recruited to the RFB, in contrast to other proteins such as RPA and Rad51 which form oligomers. We remain unable to observe Pku70 recruitment to the active RFB in wt strain, whereas Pku70 is recruited to telomeres in the same ChIP experiments (Fig. 4a and b). Similar data were obtained for the *mre11-D65N* mutant (Supplementary Fig. 2d and e). Cell synchronization has failed to provide better data. We concluded that the recruitment of Ku to the arrested fork is too transient, being quickly evicted by MRN. In support of this, it has been estimated that mammalian Ku binds DSBs in less than a second and that MRN is recruited to DSBs within 10-30 seconds; this rapid interplay between MRN and Ku occurring within second after DNA damage (Hartlerode et al. NSMB, 2015). This point has been added to the discussion section (page 10, lines 380-382). Finally, we observed that the lack of Ku in a context in which the MRN complex is functional, impacts several step of HR-mediated fork processing, including an extensive fork resection (Fig. 5d), a slower HR-mediated fork restart process (Fig. 6) and a decreased RPA and Rad51 recruitment (Fig. 7a-d). Therefore, we favor a model in which Ku is recruited to the active RFB in the wt cells. This point has been added in the discussion section (page 10, lines 376-382).

We would like to strengthen that 2DGE are performed after enrichment of replication intermediates genomic DNA extracted from an asynchronous cell population. Therefore, the intensity of the RFB appears as well enriched. Also, as mentioned above, the analysis of fork-resection by 2DGE indicates that Ku regulates the resection of DSB-free arrested forks (page 6, lines 184-188). On the principle, we cannot formally exclude that a minor fraction of the active RFB experiences a DSB to which Ku is recruited. Collectively, our genetics and molecular data do not support this scenario. However, to remain rigorous in our interpretation, we have stated in the manuscript that Ku is recruited to dysfunctional forks and avoided the statement that Ku is recruited to DSB-free arrested forks (page 7, line 236).

d) Broadly, Figure 5 addressing the impact of Ku mutation on RPA binding is the weakest part of the paper. Based on the results provided, I think the bullet point conclusion on an impact of Ku on RPA is premature. The weak fold change in the microscopic assay isn't convincing. And the IP showing a Ku-RPA assay is not strong support, given in part non-specific signal shown in Supplemental, coupled with the fact that, even with benzonase treatment, it cannot be concluded that the Ku-RPA interaction is direct. I do not say the conclusion is wrong, simply that more support it required to make it.

In the revised version, we provide data showing that RPA and Rad51 recruitment to the RFB are reduced in the *pku70-d* mutant (Fig. 7d) (page 9, lines 322-326), further supporting our microscopy studies. We have down toned our conclusion and stated in the abstract that "The mere lack of Ku impacts the processing of arrested forks, leading to an extensive resection, a reduced recruitment of RPA and Rad51 and a slower fork-restart process." The main text has been rewritten accordingly in the results section. We have added a paragraph in the discussion section to discuss the mechanism by which Ku may fine tune HR-mediated fork restart (page 11, lines 415-427).

3) It is unclear what that basis is for excluding "outliers" in the *ura3* reversion data. I understand this for truly extreme cases (e.g. where every cell has reverted), but when effects are small, only a few fold, outlier exclusion might be affecting results in a way not judgeable by the reader.

The replication slippage (or *Ura*⁺ reversion) assay starts by picking single colonies on 5FOA-containing plates which are then growth for several generations on non-selective media then plated on selective media (to select for *Ura*⁺ reversion) or not (for cell survival). Despite the work being conducted using freshly defrost strains, many strains used in this study are liable to suppressor accumulations (such as *rad50-d*, *ctp1-d* and corresponding double mutants) which are unavoidable. Suppressors may impact replication outcomes at the RFB but also cell survival. Thus, we have decided to exclude outliers from our analysis (explained in the methods section pages 14-15, lines 541-552). Most of scatter plots include from 10 up to 39 samples from independent biological replicates. We are confident that the number of samples analyzed makes our statistical analysis robust enough to support our conclusion.

4) Overall, the paper is well constructed and logical, including easy to follow figures. However, the manuscript text is very rough in many places. Several additional rounds of editing are required to make it more readable, including attention to many small grammatical issues and other points above. If necessary, consultation on English usage should be considered.

Grammatical errors were corrected.

5) It is a small point, but in my reading the term RFB is usually reserved for normal genetic elements that impair replication fork progression; I am not clear that it is common to use it as a global term to encompass all forms of replication inhibition, where many authors might use the terms replication inhibition or replication stress.

The paragraph in the introduction has been rewritten accordingly to this comment (page 3, lines 62-65).

Reviewer #3 (Remarks to the Author):

By using a genetic system that blocks a replication fork at a specific locus, the authors investigate resection and restart of terminally-arrested replication forks. Similar to what happens at DNA double-strand break, they find that MRN and Ctp1 initiate resection at arrested replication forks by removing Ku from DNA. This initial resection allows long-range resection by Exo1. Finally, they show that the lack of Ku increases resection but impairs RPA recruitment, suggesting a role for Ku in recruiting RPA to ssDNA.

The experiments are well done and the results are interesting. However there are two important points that need to be clarified/addressed:

1-The authors propose that “an initial resection mediated by MRN/Ctp1 removes Ku from terminally-arrested forks” (see abstract). However, the Mre11 nuclease activity appears to be dispensable, indicating that Ku removal is not due to a nucleolytic cleavage. In budding yeast, the lack of MRX (but not of Sae2 or Mre11 nuclease activity) increases Ku association to DSBs (Shim et al., 2010), suggesting a competition between Ku and MRX for the same DNA ends rather than a direct MRX-mediated eviction of Ku. However, the authors show that also the lack of Ctp1 (Sae2 in *S. cerevisiae*) (which should affect MRN nuclease activity and not MRN integrity) impairs Ku accumulation. These findings leave me confused because if Mre11 nuclease is not involved, the lack of Ctp1 should mimic the result obtained with the Mre11 nuclease defective mutant. This is an important point that needs to be clarified. Is Ku association impaired in *rad50-d*, *ctp1-d* and *mre11-D65N* cells? Is Mre11-D65N nuclease defective in vitro? Can the authors test other *mre11* nuclease defective allele(s)?

In the revised version, we provide new ChIP experiments, after optimizing our protocol (changes in the protocol have been introduced in the Methods section, page 14, lines 517-528). We have also introduced an internal control which is the recruitment of Pku70-HA to telomeres (Fig. 4a and b, Supplementary Fig. 2d and e). In the absence of Rad50, Pku70 accumulated upstream from the active *RTS1*-RFB. A similar recruitment, although to a lesser extent, was observed in *ctp1-d* cells (Fig. 4a) (page 7, lines 244-251). We have analysed the *mre11-D65N* mutant and found no accumulation of Pku70 at the active RFB whereas it is recruited to telomeres (Supplementary Fig. 2 d-e). We have analysed another nuclease dead *mre11* mutant (*mre11-H134S-Myc*, send by the lab of P. Russell) and found no defect in downstream and upstream RFB-induced replication slippage in this mutant (Supplementary Fig. 2a and b). We haven't tested in vitro the nuclease activity of these mutated forms of Mre11. However, the Mre11-D65N and Mre11-H134S are equivalent to the well-characterized *S. cerevisiae* Mre11-D56N and Mre11-H125N, respectively, which have been shown to have negligible exonuclease activity and a complete loss of endonuclease activity in vitro (Krogh et al. Genetics 2005, Moreau et al. MCB 1999). These data further support that the nuclease activity of Mre11 is dispensable to promote Ku eviction from arrested forks and initiate end-resection. We have proposed in the discussion that the role of MRN and Ctp1 in promoting fork-resection and restart involved a structural role rather than a nucleolytic role (page 11, lines 383-393).

By analyzing the resection of DSBs in fission yeast, Langerak and colleagues have reported that end-resection requires MRN and Ctp1 but not the nuclease activity of Mre11 (using the *mre11-H134S*

mutant). In contrast, Ku removal requires MRN, Ctp1 and the nuclease activity of Mre11. Therefore, the requirement of the Mre11 nuclease activity in Ku removal is different at DSB-free arrested forks and DSBs. We have added this point to the discussion (page 10, lines 383-387).

It has not been shown in vitro that fission yeast Ctp1 stimulates the nuclease activity of Mre11 in vitro. Thus, we have rewritten the sentence “The endo- and exo-nuclease activities of Mre11, stimulated by Sae2/Ctp1, are not strictly required to DNA end resection at “clean” DSBs” in “The endo- and exo-nuclease activities of Mre11, stimulated by Sae2, are not strictly required to DNA end resection at “clean” DSBs” (page 3, lines 92-95).

We agree that the question of competition between Ku and MRN for DSBs binding versus the alternative possibility that Ku binds DSBs and is then evicted by MRN is a longstanding question. We agree that the fact that we observed no Ku recruitment to the RFB in wt cells (Fig. 4a) may favor the second hypothesis. However, we report that the sole lack of Ku impacts replication outcomes at the RFB, including extensive fork resection, a reduced RPA and Rad51 recruitment and a slow HR-mediated fork restart process. The mere lack of Ku (when MRN and Ctp1 are functional) impacts the processing of arrested forks, leading to an extensive resection, a reduced recruitment of RPA and Rad51 and a slower fork-restart process (abstract). Thus, we believe that collectively, our genetics and molecular data favors a model in which Ku is recruited to arrested forks and then evicted by MRN and Ctp1. We added in the discussion section: “Possibly, MRN-Ctp1 initiates fork-resection to create a substrate less favorable to Ku binding¹⁹. However, the lack of Ku impacts replication and recombination outcomes at the RFB, suggesting that Ku binds arrested forks even if the MRN-Ctp1 axis is functional. Thus, we propose that Ku is recruited early, likely on the reversed arm, and is then quickly evicted by MRN-Ctp1. In support of this, a rapid interplay, occurring within seconds after DNA Damage, have been reported between mammalian Ku and MRN in mammals” (page 10, lines 377-382).

2-The authors propose that Ku helps the recruitment of RPA to ssDNA. To my knowledge there are no evidence that the lack of Ku impairs RPA association at DSBs. Rather, *ku-d* cells hyperactivate a Mec1-dependent checkpoint and therefore they should allow RPA loading onto ssDNA. In addition, since Ku binds the double-stranded DNA, it is no clear to me how Ku could help the association of RPA to ssDNA. The authors should use another assay (ChIP?) to measure quantitatively the association of RPA in Ku mutants because the difference in RPA foci formation between *ku-d* and wt is really very subtle.

In the revised version, we provide data showing that RPA and Rad51 recruitment to the RFB are reduced in the *pku70-d* mutant (Fig. 7d) (page 9, lines 322-326), further supporting our microscopy studies. We have down toned our conclusion and state in the abstract that “The mere lack of Ku impacts the processing of arrested forks, leading to an extensive resection, a reduced recruitment of RPA and Rad51 and a slower fork-restart process”. The main text has been rewritten accordingly in the results section. We have added a paragraph in the discussion section to discuss the mechanism by which Ku may fine tune HR-mediated fork restart (page 11, lines 415-427).